# Indicator-to-impact links to help improve agricultural drought preparedness in Thailand

Maliko Tanguy[1]*, Michael Eastman[1,2], Eugene Magee[1], Lucy J. Barker[1], Thomas Chitson[1], Chaiwat Ekkawatpanit[3], Daniel Goodwin[4,5], Jamie Hannaford[1,6], Ian Holman[5], Liwa Pardthaisong[7], Simon Parry[1], Dolores Rey Vicario[5], Supattra Visessri[8,9]

[1] UK Centre for Ecology & Hydrology (UKCEH), Wallingford, United Kingdom
[2] Met Office, Exeter, United Kingdom
[3] Department of Civil Engineering, King Mongkut's University of Technology Thonburi, Bangkok, Thailand
[4] School of Social Sciences, University of Tasmania, Australia
[5] Cranfield University, Cranfield, United Kingdom
[6] Irish Climate Analysis and Research UnitS (ICARUS), Maynooth University, Ireland
[7] Department of Geography, Faculty of Social Sciences, Chiang Mai University, Chiang Mai, Thailand
[8] Department of Water Resources Engineering, Faculty of Engineering, Chulalongkorn University, Bangkok, Thailand
[9] Disaster and Risk Management Information Systems Research Unit, Chulalongkorn University, Bangkok, Thailand

*Correspondence to*: Maliko Tanguy (malngu@ceh.ac.uk)

**Abstract.** Droughts in Thailand are becoming more severe due to climate change. Developing a reliable Drought Monitoring and Early Warning System (DMEWS) is essential to strengthen a country's resilience to droughts. However, for a DMEWS to be valuable, the drought indicators provided to stakeholders must have relevance to tangible impacts on the ground. Here, we analyse drought indicator-to-impact relationships in Thailand, using a combination of correlation analysis and machine learning techniques (random forest). In the correlation analysis, we study the link between meteorological drought indicators and high-resolution remote sensing vegetation indices used as proxies for crop-yield and forest-growth impacts. Our analysis shows that this link varies depending on land use, season, and region. The random forest models built to estimate regional crop productivity allow a more in-depth analysis of the crop-/region-specific importance of different drought indicators. The results highlight seasonal patterns of drought vulnerability for individual crops, usually linked to their growing season, although the effects are somewhat attenuated in irrigated regions. Integration of the approaches provides new detailed knowledge of crop-/region-specific indicator-to-impact links, which can form the basis of targeted mitigation actions in an improved DMEWS in Thailand, and could be applied in other parts of Southeast Asia and beyond.

## 1 Introduction

Droughts are one of the costliest natural hazards worldwide (FAO, 2021). Their frequency and duration are expected to increase in many parts of the world due to climate change (IPCC, 2021, 2022; WBG & ADB, 2021). Over the past decades, Thailand has already seen a rise in impacts from a warming world, experiencing an increasingly unpredictable weather, with an

alternation of droughts and floods on a two-three year cycle (Ikeda & Palakhamarn, 2020), causing a wide range of impacts. This trend is expected to intensify further in the near future in South-East Asia as highlighted by Hariadi et al. (2023).

One notable recent example is the severe 2020 drought, which was driven by a shorter monsoon period and a strong El Niño event (CFE-DMHA, 2022). The drought caused impacts in water supply, water quality, crop production and the economy, with an economic loss of THB46 billion (US$1.4 billion, £1.1 billion; Sowcharoensuk & Marknual, 2020). Other notable recent droughts include the 2005 event, in which 11 million people in 71 (out of 77) provinces were affected by water shortages; the 2008 event where over 10 million people in rural areas were affected (Ikeda & Palakhamarn, 2020); and the 2015-2016

event, which affected the upper-middle part of the country most, and was so severe that at the Ubolrat dam, in Northeastern Thailand, steps were taken to use "dead storage" (i.e., the last 1% in the bottom of the reservoir; CFE-DMHA, 2022). Overall, the National Disaster Relief Centre estimates that drought events between 1989 and 2017 caused more than B19.1 billion (US$0.6 billion, £0.5 billion) of damage to the Thai economy, with average annual economic damages of almost THB0.6 billion per year (US$20 million, £16 million; NESDC, 2021).

One sector particularly affected by droughts in Thailand is agriculture (Yoshida et al., 2019); in particular, rice, corn, and other cash crops periodically suffer economic losses (Ikeda & Palakhamarn, 2020). Thailand is currently the second largest rice exporter in the world (OECD, 2020), and rice fields utilises 70% of Thailand's total water supply (ICID, 2020). Thailand is also the second biggest sugar exporter, and the 2020 drought resulted in a 28% fall in production (Thammachote & Trichim, 2021). However, drought risk is also moderated or exacerbated by human activities. Areas with water reservoirs and extensive

irrigation facilities are more resilient and impacted less by droughts than rainfed agriculture and areas without reservoir storage. In the Northeast, higher water demand for rice cultivation during the dry seasons, combined with limited irrigation infrastructure, exacerbates water scarcity (CFE-DMHA, 2022).

Thirty percent of Thailand's population work in agriculture, and drought threatens their income and poses food security issues. Given this considerable impact that droughts have on Thai society, and the expectation of a worsening in the coming years and

decades, there is an urgent need to improve preparedness and resilience of the country to droughts (UNDRR & ADCP, 2020). This also aligns with the priorities of the UNDRR's Sendai Framework for Disaster Risk Reduction 2015-2030, which aims to achieve the substantial reduction of disaster risk and losses in lives, livelihoods and health and in the economic, physical, social, cultural and environmental assets of persons, businesses, communities and countries over the next 15 years (UNDRR, 2015). One important aspect of improving drought resilience lies in enhancing the Drought Monitoring and Early Warning

(DMEW) capabilities of the country, in order to detect droughts in their early stages such that proactive mitigation strategies can be implemented (Bachmair et al., 2016a).

According to the World Meteorological Organization (WMO), drought can be defined as a prolonged dry period in the natural climate cycle (WMO, 2014). Since Wilhite and Glantz (1985), drought has commonly been categorised into various types often differentiating between meteorological, hydrological, and soil moisture (or agricultural) droughts, alongside various

others. Many drought indices have been developed for drought monitoring purposes for these different types of drought over the past decades (Lloyd-Hughes, 2014). In this paper, we consider a drought to be a period drier than normal for that time of

year at a given location and is distinct from the impacts it causes. Droughts can occur without causing any impacts, in which case they are not a concern for water managers or water users.

Drought impacts (e.g. crop failure, water quality issues, etc.) are what stakeholders are interested in from a DMEW point of view. However, impact data is scarce, and generally not routinely monitored. There are significant challenges in collecting and monitoring drought impacts, including the visibility of drought impacts, which can be diffuse, delayed and non-structural (e.g. in comparison to the impacts of flooding). Nevertheless, while impact research is inherently challenging, it is also pivotal to drought management. While rainfall or river flow deficits can help track drought evolution, ultimately it is the impacts of drought which are of greatest importance for water managers and other stakeholders. Numerous international initiatives have highlighted that information on drought impacts is the key 'missing piece' of drought monitoring and forecasting (e.g. Bachmair et al., 2016a), and some effort has been invested in collating drought impact data at national or international scale (e.g. Europe: European Drought Impact report Inventory (EDII), Stahl et al. (2016); U.S.: Drought Impact Reporter (DIR), Smith et al. (2014)). In an age where there have been huge advances in real-time hydrometeorological monitoring, better prediction of impacts would be the single greatest practical advance in paving the way for improved drought resilience. Understanding the link between drought indicators and impacts is an essential first step to achieve this goal (Bachmair et al., 2016a).

Some of the most commonly used indices in operational DMEWS are the meteorological standardised indices such as Standardised Precipitation Index (SPI, McKee et al., 1993) and Standardised Precipitation-Evapotranspiration Index (SPEI, Vicente-Serrano et al., 2010). However, these indicators based purely on meteorological status are not always well correlated to drought impacts (Bachmair et al., 2018), as impacts often occur when precipitation deficits have propagated through the hydrological cycle to deficits in soil moisture or river flows, for example. Moreover, precipitation deficit is likely to cause more impacts in water-limited regions than in regions with abundant water, though water management practices can counteract this effect to a certain extent. Drought indices are only meaningful to decision makers if the relationship to drought impacts is known, i.e. understanding the type and magnitude of impacts that can be expected for different drought index values. For regions where drought impact data are available, the relationship between drought indices to drought impacts can be studied (e.g., Bachmair et al., 2016b; Parsons et al., 2019; Wang et al., 2020). Where drought impact data are not readily available, remote sensing vegetation indices (VIs) can provide a proxy for drought impacts on vegetation.

VIs are commonly used to monitor the impacts of drought on vegetation. The Normalised Difference Vegetation Index (NDVI) is one of the most established and widely used VIs (Tucker, 1979). It exploits the sharp increase in vegetation reflectance across the red and near-infrared (NIR) regions of the electromagnetic spectrum, known as the 'red-edge', to detect photosynthetically active plant material and infer plant stress. However, the Vegetation Condition Index (VCI), a pixel-based normalization of NDVI, offers a more robust indicator for seasonal droughts by minimising spurious or short-term signals and amplifying long-term trends (Anyamba & Tucker, 2012; Liu & Kogan, 1996). VCI has been widely used and has proved to be effective in monitoring vegetation change and signalling agricultural drought (e.g. Jiao et al., 2016). The Vegetation Health Index (VHI) is a composite index that combines the VCI and Temperature Condition Index (TCI) – a pixel-based normalisation

of the Land Surface Temperature (LST) – and is also commonly used to monitor vegetation stress and drought conditions (Kogan, 1997). VHI incorporates the effect of temperature and is therefore more suitable for monitoring the effect of drought in species more sensitive to concurrent water and heat stress. VHI has been successfully used worldwide to monitor vegetation stress and drought conditions (e.g. Jain et al., 2009; Singh et al., 2003; Unganai & Kogan, 1998). Note that these VIs are relative indices that compare current conditions to the long-term average to measure vegetation health, and therefore are dependent on the environmental and climatic conditions of the study area. As such, they should be used in conjunction with information on the drought hazard situation to distinguish between drought and different hazards on vegetation (e.g. disease, floods, anthropogenic impacts, etc.).

In addition to their use as drought indicators as discussed above, VIs are often used as proxies for agricultural drought impacts. The relationship between crop yield and VIs varies by crop type and location but has been shown to be strong in many locations. For example, strong correlations were found between VIs and crop yield in North America (e.g. maize in Bolton & Friedl, 2013; winter wheat, sorghum and corn in Kogan et al., 2012), South America (e.g., white oat in Brazil in Coelho et al., 2020), Europe (e.g., maize in Germany in Bachmair et al., 2018; cereals in Spain in García-León et al., 2019), Asia (e.g., sugarcane in India in Dubey et al., 2018), the Middle East (e.g., paddy rice in Iran in Shams Esfandabadi et al., 2022), Africa (e.g., millet and sorghum in the Sahelian region in Maselli et al., 2000), and Australia (e.g., wheat in Smith et al., 1995).

Data science and machine learning is a fast-moving field and is increasingly being used for the study of environmental science, though still in its infancy (Blair et al., 2019). Random Forest (RF) models have been used to link drought indicators to drought impacts (e.g. Bachmair et al., 2016b), including drought impact forecasting with relative success (Hobeichi et al., 2022; Sutanto et al., 2019). These emerging techniques within the field of DMEW offer great potential to move from simply monitoring droughts using indices to drought impact estimation, which would revolutionise the early warning aspect of drought mitigation, enabling action to be taken before impacts occur.

Despite the significance of droughts in Thailand, few previous studies have analysed the link between drought indices and drought impacts in the country. Thavorntam et al. (2015) and Thavorntam and Shahnawaz (2022) looked at links between SPI and VIs, but only at four test sites in the North-East of Thailand. Prabnakorn et al. (2018) and Khadka et al. (2021) have both focused on the drought-prone Mun River Basin situated in the North-East of Thailand; both studies find that SPEI shows a good correlation to crop yield. However, no previous study has looked comprehensively at drought indicator-to-impact links at a national scale in Thailand, and to our knowledge none has used machine learning techniques to estimate drought impacts in the country.

The ambition of this paper was to fill the gap in the literature on studies investigating the links between drought indicators and impacts at a national scale in Thailand. Specifically, we focused on agricultural drought impacts, considering different crops and seasons, and compared the relative utility of traditional statistical methods at high resolution (i.e. remote sensing data at provincial scale) vs. lower resolution sectoral-specific analyses (i.e. applying machine learning approaches to regional/provincial yield data), to inform improved approaches for national DMEW.. The overall aim was to support agricultural drought management and inform targeted action/policy by water resource managers. To that end, this paper

evaluates how relationships between droughts indices and impacts vary according to time of year, index, accumulation period length and location in Thailand. The approach presented is relevant internationally and could be replicated in other parts of the world to improve the management of agricultural droughts and their impacts.

## 2. Data and methods

### 2.1 Study area

Thailand is located between 5°30' and 20°30'N latitudes and between 97°30' and 105°30'E longitudes, and has an area of 51 million ha, from which 46.5% is agricultural area (77% of which is rainfed). Paddy fields covers 46% of that cultivated area, with around 30% being irrigated (OAE, 2022).

Most of the country experiences distinct wet and dry seasons, except some parts of the southern region, which experience a wet and humid climate throughout the year. The average annual rainfall of the whole country is about 1,700 mm ranging from 1,200 mm in the north and central plain up to 2,000 - 2,700 mm in the western part of the south and the eastern part of the country (ICID, 2020).

Droughts often occur in two distinct periods: between June and September as a consequence of a delay in the onset of rainfall, or due to low precipitation during the dry season between October and May. The occurrence of drought in Thailand is increasingly associated with the El Niño-Southern Oscillation (ENSO) cycle, which brings drier-than-average rainfall conditions (UNDRR & ADCP, 2020).

Figure 1 shows the six regions and 77 provinces that were used in our analysis. Provinces (*changwat*) are the primary local government unit in Thailand. The regions do not have an administrative character, but are commonly used for geographical and scientific purposes (e.g. Martin & Ritchie, 2020; Sanoamuang & Dabseepai, 2021). The dominant land cover for each province is shown in Fig. SF1 in the Supplementary Information (SI).

In addition to these regions and provinces, in the results and discussion sections, we use 'the North' to refer to the area encompassing regions N, NE, C, W and E, as opposed to 'the South' comprising only region S.

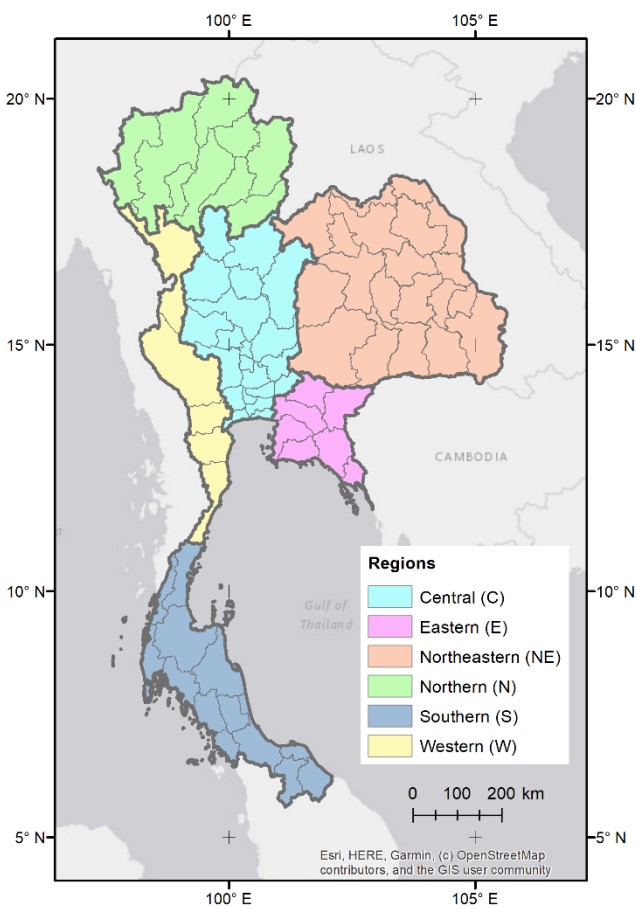

**Figure 1: Map of Thailand showing provinces and regions used in this study. The provinces are the smaller areas shown within each of the coloured regions.**

**Regional differences:**

The Northeastern region (NE) consists mainly of the dry Khorat Plateau. Unlike the more fertile areas of Thailand, the NE has a long dry season, and much of the land is covered by sparse grasses. The main crops cultivated in this region are glutinous rice (two harvests), cash crops such as sugar cane and cassava, and to a lesser extent rubber. This region is the most prone to drought (LePoer, 1987), and as such, is particularly vulnerable to agricultural droughts as highlighted by several studies (e.g. Mongkolsawat et al., 2001; Sa-nguansilp et al., 2017; Wijitkosum, 2018).

The Northern region (N) is a mountainous region, and the most forested region of Thailand. Although it has suffered from extensive deforestation due to agricultural expansion over the past few decades, there has been some reforestation in recent years (RFD, 2022). Many dams and irrigated croplands are situated in this region.

The Western (W) region is characterised by high mountains and steep river valleys. Western Thailand hosts much of Thailand's less-disturbed forest areas. The region is home to many of the country's major dams.

The Eastern (E) region is characterised by short mountain ranges alternating with small basins of short rivers which drain into the Gulf of Thailand; fruit is a major component of agriculture in the area.

The Central (C) region is a natural self-contained basin often termed "the rice bowl of Asia". A complex irrigation system and fertile soil supports the cultivation of rice paddies. It is the most densely populated region of Thailand, with Metropolitan Bangkok on its southern edge (LePoer, 1987).

The Southern region (S) is part of a narrow peninsula, and is distinctive in climate, terrain, and resources. This region is characterised by North-south mountain barriers, tropical forest, and the absence of large rivers. It is the wettest region in Thailand, and is not generally considered to suffer from drought impacts (LePoer, 1987).

## 2.2 Data

Table 1 lists the data used in this study, with details on the type of data, spatial resolution, temporal resolution, period available, post-processing applied in this study and reference.

From these datasets, the following drought indicators were calculated:

- **Standardised meteorological indicators**: Standardised Precipitation Index (SPI, McKee et al., 1993) and Standardised Precipitation Evapotranspiration Index (SPEI, Vicente-Serrano et al., 2010) for accumulation periods of 1-6, 9, 12, 18 and 24 months. For the SPI, the data was fitted to a gamma distribution, whereas for SPEI, a generalised logistic distribution was used, as recommended by the original authors.
- **Vegetation Indices (VIs) from remote sensing**: Vegetation Condition Index (VCI), Temperature Condition Index (TCI) and Vegetation Health Index (VHI) were calculated on a monthly time-step following Bachmair et al. (2018) methodology, which is detailed in the Supplementary Text 1 (ST1) of the SI.

In this study, when the word 'indicators' is used on its own, we refer to both meteorological indicators (SPI and SPEI) and VIs (VCI, TCI and VHI).

Annual crop yield data (OAE, 2021) are used as a measure of agricultural drought impacts. Although drought is not the only factor that can cause crop yield departure, Venkatappa et al. (2021) have shown that it is the main driver of crop loss in Thailand.

**Table 1: Details of the datasets used in this study**

| Dataset | Type of data | Variable | Spatial reso-lution | Tem-poral reso-lution | Period avai-lable | Post-processing | Reference |
|---|---|---|---|---|---|---|---|
| APHRODITE (Asian Precipitation - Highly- Resolved Observational Data | Gridded data interpolated from ground observations | Precipitation (P) | 0.25deg | Daily | 1998-2015 | Used to calculate the Standardised Precipitation Index (SPI). | Yatagai et al. (2012) |

| | | | | | | | | |
|---|---|---|---|---|---|---|---|---|
| Integration Towards Evaluation) | | | | | | | | |
| MOD16A2 product from MODIS | Gridded data from remote sensing / modelled data | Potential Evapotrans-piration (PET) | 500m | 16-day | 2000-2020 | The Climatic Water Balance (CWB) is calculated as P – PET. CWB used to calculate the Standardised Precipitation Evapotranspiration Index (SPEI). | Running et al. (2017) |
| MCD12Q1 product from MODIS | Gridded data from remote sensing | Land cover map | 500m | Annual | 2000-2015 | Land cover map and dominant land cover for each province are shown in Figure SF1 of the SI. Used to create cropland and forest masks. | Friedl and Sulla-Menashe (2019) |
| MOD13A1 and MYD13A1 products from MODIS | Gridded data from remote sensing | Normalized Difference Vegetation Index (NDVI) | 500m | 16-day | 2000-2020 | NDVI and LST masked using crop and forest masks, before aggregation at province level. Used to calculate the Vegetation Condition Index (VCI), Temperature Condition Index (TCI) and Vegetation Health Index (VHI). VCI for crops de-trended to remove effect from technological advances. | Didan (2015a, 2015b) |
| MOD11A2 product from MODIS | Gridded data from remote sensing | Land Surface Temperature (LST) | 1km | Monthly | 2000-2022 | | Wan et al. (2015) |
| Crop yield data | Yearly time series per crop and province | Crop yield | Pro-vince level | Annual | 1984-2019 | Main crop in each province identified. Time series de-trended to remove effect from technological advances. | Office of Agricultural Economics (OAE, 2021) |

**Spatial and temporal aggregation**

To derive the meteorological indicators, we first averaged the meteorological variables (precipitation and PET) for each province and then calculated the standardised indicators based on the province-averaged time series. For VIs, we first derived them at the pixel level for the entire country, and then used a land cover map to differentiate between forest and crop-covered pixels. We then calculated province-level VIs averages separately for forest and crops, using the corresponding land cover mask. We used monthly time series for most of our analysis, with the exception of the comparative analysis between VIs and

crop yield (described further in section 2.3.1.1) where VIs were averaged over the growing season for each crop.

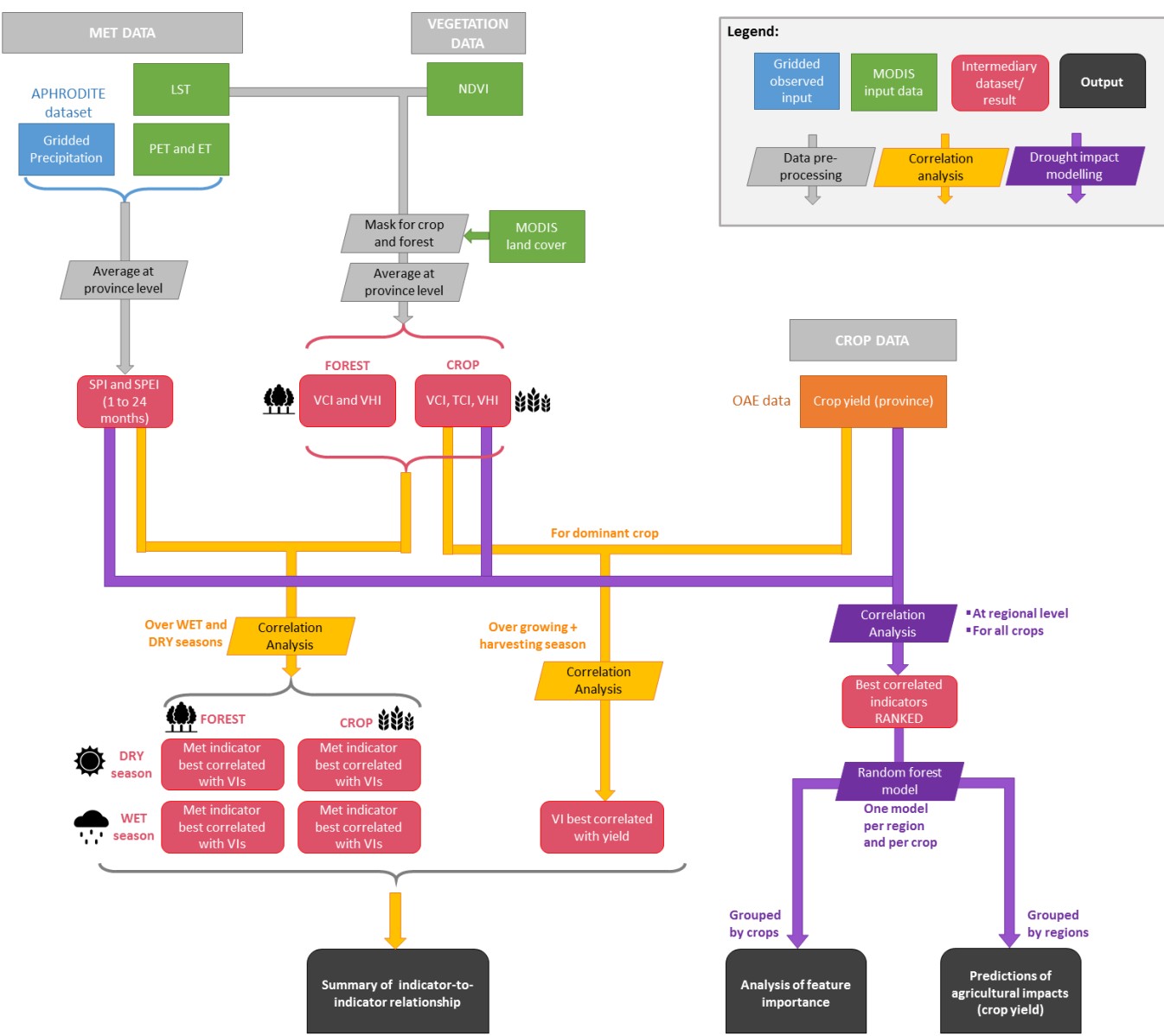

**Figure 2: Schematic diagram of the steps involved in this study**

## 2.3 Methods

Figure 2 shows a schematic representation of the methodological steps involved in this study.

### 2.3.1 Indicator-to-Indicator Correlation analyses

For the correlation analysis, data was used at the finer province resolution.

### 2.3.1.1. VIs vs. crop yield

Firstly, a correlation analysis was performed between the vegetation indices (VIs) and the crop yield data. This was done to investigate whether the VIs could be used as a proxy for agricultural impacts as spatially distributed data on crop areas were unavailable to accompany the yield data. VIs were masked using the land cover data to ensure that only areas covered by cropland were considered. The cropland-masked monthly VIs were then averaged to the province level, the time series filtered to only include the growing season of the spatially-dominant crop within each province, and an annual average was taken. The growing season was taken from Lacombe et al. (2017) for Cassava, FAO GIEWS Country Brief (FAO, 2021) for Paddy rice, Arunrat et al. (2022) for Corn and FFTC (2015) for Longan. The annual time series for the VIs for each province was correlated with the yield of the dominant crop for that province using a Pearson correlation (Pearson, 1920). The Pearson correlation was selected since it estimates the strength of normalised covariance between two variables, allowing for insight into how closely related the two variables are.

### 2.3.1.2. Meteorological drought indicators vs. VIs

The Pearson correlation was also used to compare the standardised indicators (SPI and SPEI) and vegetation indices (VCI and VHI) for both forest areas and cropland, where the crop-masked vegetation indices were treated as a proxy for the agricultural impact. This approach was used to investigate the effect of meteorological conditions on crops and forests, and identify the most relevant indicators from a drought monitoring perspective. Monthly crop-masked VCI values were regressed against time using linear regression, and the residuals used to remove linear trends, accounting for increased biomass from developments in agricultural technology and practices. The analysis was done spatially, making use of province averaged indicators, and temporally, by splitting the time series into wet and dry seasons. Whilst the specific months of these seasons varies across the country, a general approach was taken with the wet season being May to October and the dry season November to April, inclusive. Correlation coefficients were calculated between standardised meteorological indicators for all given accumulation periods and the VIs. For each VI and province, the standardised meteorological indicator with the largest magnitude correlation was identified, and critical values were calculated by accounting for autocorrelation using Pyper and Peterman (1998)'s methodology.

To check how much difference there is between SPI and SPEI, and verify that they are different enough to justify using both indices in our analysis, we compared the two indicators to determine how much of SPEI can be explained by SPI over the whole period, each season and each accumulation period, details of which are given in ST2 of the SI.

### 2.3.2 Simulating Crop Productivity

Regional Random Forest (RF) models were used to predict agricultural impacts (crop yield). RF Regression is a machine learning algorithm that combines predictions from multiple decision trees to make a more accurate prediction than a single tree. The analysis was carried out at regional level by aggregating all provincial data to the regional level, as data was too scarce at the provincial level to be able to train the models at that higher resolution.

As input data to the models, we used SPI, SPEI, VCI, and TCI for each individual month separately. Note that VHI was not used here as it is a combination of VCI and TCI, and is therefore strongly correlated to both. Accumulation periods of 1-6, 9, 12, 18 and 24 months, were used for SPI and SPEI. All input data were first regressed against time, and the residuals used as input to the random forest models to account for linear trends. Annual crop yield data for a range of individual crops were used to train the models and evaluate them.

First, a correlation analysis of all input data against each crop's annual yield was carried out with the objective of ranking the indicators in order of highest to lowest correlation with crop yield. All indicators were split by month (i.e. all the Januarys lumped together, all the Februarys, and so on), and the correlations were ranked by p-value.

In a second step, we built the feature set by adding features (i.e. indicators) in order of ascending correlation p-value, whilst maintaining all Variance Inflation Factor (VIF) values below 5 (to minimise multicollinearity). This means that for strongly correlated input variables, only the variable with the strongest correlation with crop yield was used to build the model.

In the final step, we built the forests to predict crop yield. A total of 38 individual RFs were built for each combination of crop and region using the six regions shown in Figure 1, and seven crops – Cassava, Corn S1 (March-October), Corn S2 (November-February), Mixed Corn (Corn S1 + Corn S2), Paddy rice, Second rice and Longan. Only combinations that had more than 50 samples (province yield-year combinations) were used, and as a result, Corn S1, Corn S2 and mixed Corn were removed from region S, and Corn S2 was removed from region E. Figure 3 shows a schematic representation of the steps involved to build the RF models.

Due to the considerable number of RFs trained and evaluated, the number of trees within each RF was selected using an automated process by evaluating the mean squared error for RFs consisting of 50, 100, 1000, 10000 trees. The number of trees that resulted in the lowest mean squared error was used to train the final model for each region-crop combination. To enable parameters to be estimated on the full dataset, estimation of the optimal number of trees, and training of the final model, was performed using 5-fold cross validation.

Finally, using these models, we investigated the relative importance of features in explaining the variance in crop yield. The average decrease in Gini Impurity resulting from the exclusion of a certain feature can provide insights into its relative importance for simulating the target variable. In this case, indicators with relatively high decreases in Gini Impurity resulting

from their exclusion were considered important for the simulation of the productivity of the crop in question. While RFs were built to predict crop yields, the main focus of our study was their use to study feature importance to identify monitoring priorities for different regions and crops.

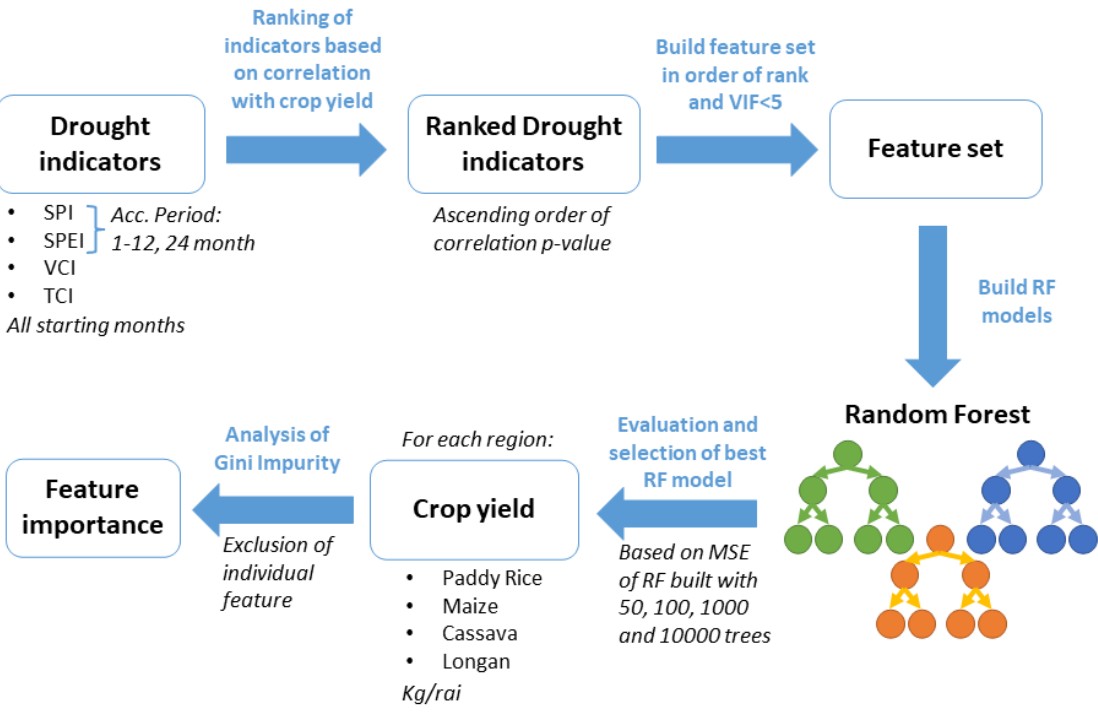

**Figure 3: Schematic representation of the steps involved to build RF models and associated analysis in this study.**

## 3. Results

### 3.1 Correlation analysis: Indicator-to-indicator

**3.1.1 VIs vs. crop yield**

In most provinces, we found that VCI is positively correlated to crop yield for the dominant crop in that province, and in the majority of cases, that correlation is statistically significant ($p \leq 0.05$) (See Fig. 4a). In thirteen provinces (out of 77), VCI is negatively correlated to crop yield (provinces in blue in Fig. 4a), which suggests VCI is not directly linked to crop yields in these provinces and may not be suitable as a proxy for agricultural impacts. In the most northern provinces, the land cover is
285 highly dominated by dense forest (Fig. SF1), and the limited crop area has a mixture of crops which might explain these poor relationships.

VHI is negatively correlated to yields in more provinces than VCI, but has stronger correlation than VCI in some provinces (Fig. 4b). Figure 4d shows the VI best correlated to crop yield (for dominant crop) in each province. For more than 90% of the provinces, at least one of the VIs is positively correlated to crop yield. Note that in some provinces, the dominant crop –
290 especially provinces in W, C and E regions – accounts for less than 50% of the total cultivated area (Fig. 4c). This can introduce significant noise in the data, and therefore these results should be treated with caution and be considered as a general indication that VIs are a reasonable proxy for crop yield, rather than an absolute validation. In some cases, there is no obvious reason as to why the correlation is very different between two neighbouring provinces which share similar topography, land cover, climatology and dominant crop type. However, differences in irrigation or agricultural practices, or in the outbreak of pests
and diseases, could be contributing factors. Exploring these factors in future research may provide insights into the observed differences in correlations. Crop yield at field scale or a high-resolution land cover map which includes information on crop type would be needed to carry out a robust validation, but in the absence of such data, we consider that the strong correlation between VIs and crop yield found in most provinces provide enough confidence to utilise VIs as a reasonable proxy for crop yield in subsequent analysis in this paper.

The following analysis focusses on VCI to simplify the messaging, but the equivalent plots for VHI can be found in the SI. Note that for this analysis, TCI was not considered, as its effect is implicit within VHI as described in Section 2.2 above.

Note that we also use VIs as proxy for forest growth in the following analysis, but we had no verification data to validate this assumption. However, VIs have shown strong links to forest health and drought impacts in previous studies (e.g. Byer & Jin, 2017; Torres et al., 2021). Therefore, we consider that the assumption that VIs are good proxies to drought impacts on forest
to be reasonable.

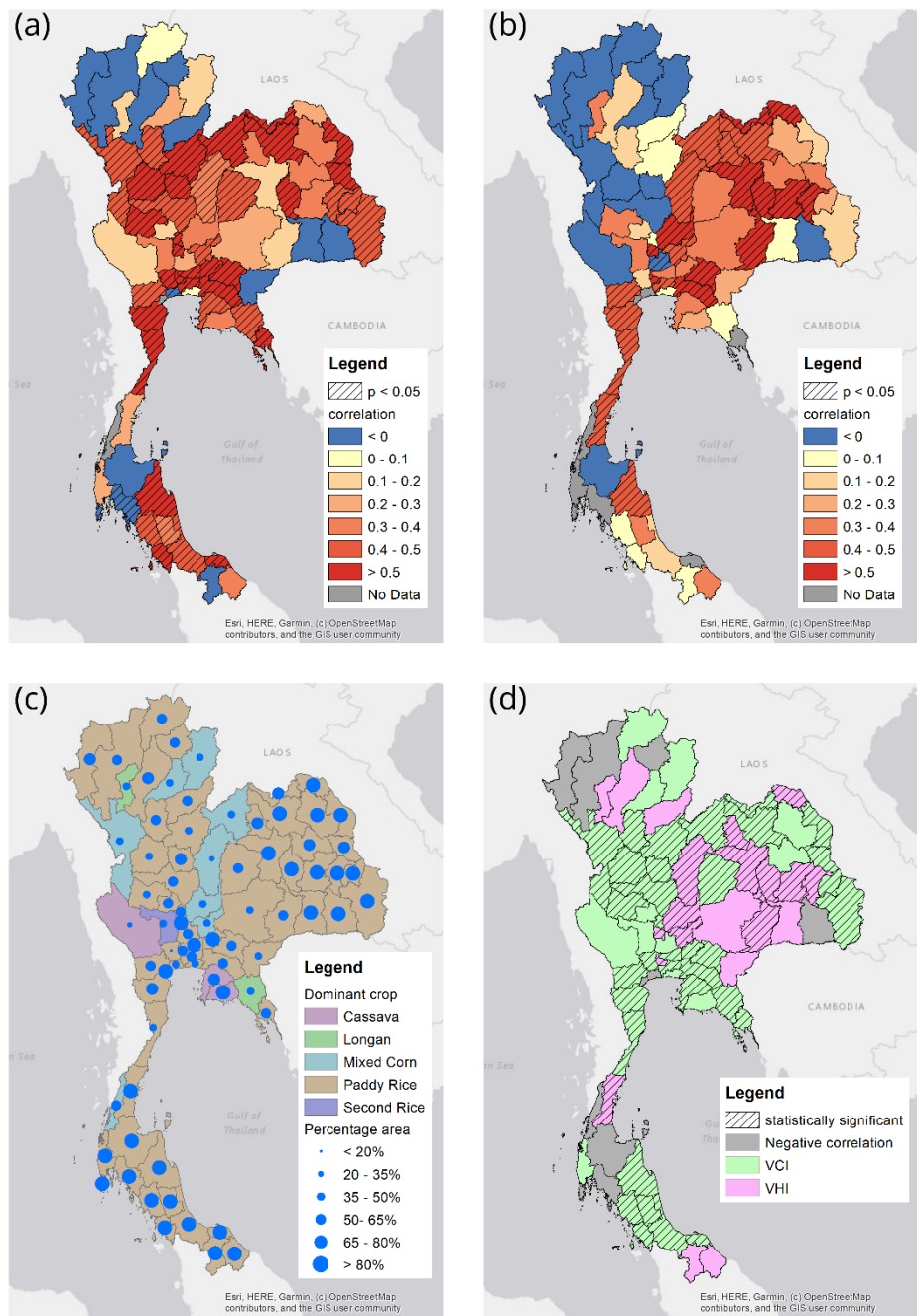

**Figure 4: (a) Correlation between VCI and crop yield for dominant crop in each province; (b) Correlation between VHI and crop yield for dominant crop in each province, (c) map of dominant crop in each province and the percentage area of said crop over the total crop area in each province; and (d) map of VI best correlated with crop yield for dominant crop in each province.**

### 3.1.2 Meteorological indicators vs. VIs

Meteorological drought indicators were then correlated with VIs to assess the effect of meteorological conditions on crops and forests and identify the most relevant drought indicator for impacts on crops and forests. The analysis was divided between dry and wet seasons.

### 3.1.2.1 Dry season

Figure 5 shows the strongest correlation for all the combinations of meteorological indicators vs. VCI for the dry season (Fig.5a and b) with the corresponding meteorological indicator (Fig. 5c and d), for crops (Fig 5a and c) and for forest (Fig 5b and d). Strong and statistically significant correlations can be seen for most provinces in the North. Correlations are higher for crops than for forest.

For crops, we find high correlations between VCI and SPEI of relatively short accumulation period during the dry season, suggesting that short droughts affect crops most. The fact that SPEI is generally more highly correlated to crop production than SPI highlights the important link between the evaporative demand and impact on crops.

For forests, we observe a very clear North-South split, with positive correlations in the North and negative in the South. A positive correlation between VCI and the meteorological indicators suggests that a deficit in water availability (as indicated by negative SPI or SPEI) leads to a decline in vegetation growth (reduced VCI). In contrast, a negative correlation suggests that such a deficit leads to an increase in vegetation growth. This second scenario may seem counterintuitive, but it can occur in energy-limited environments where water is not the limiting factor. In such cases, short duration droughts (i.e., periods drier than usual for the time of year) can stimulate increased vegetation growth, as droughts in energy-limited environments are often associated with increased radiation (i.e. energy) due to decreased cloud cover. This is discussed further in section 4.3. Except in the South, the best correlated accumulation period is generally longer for forest than for crops. Impacts on forests happen during longer droughts than for crops.

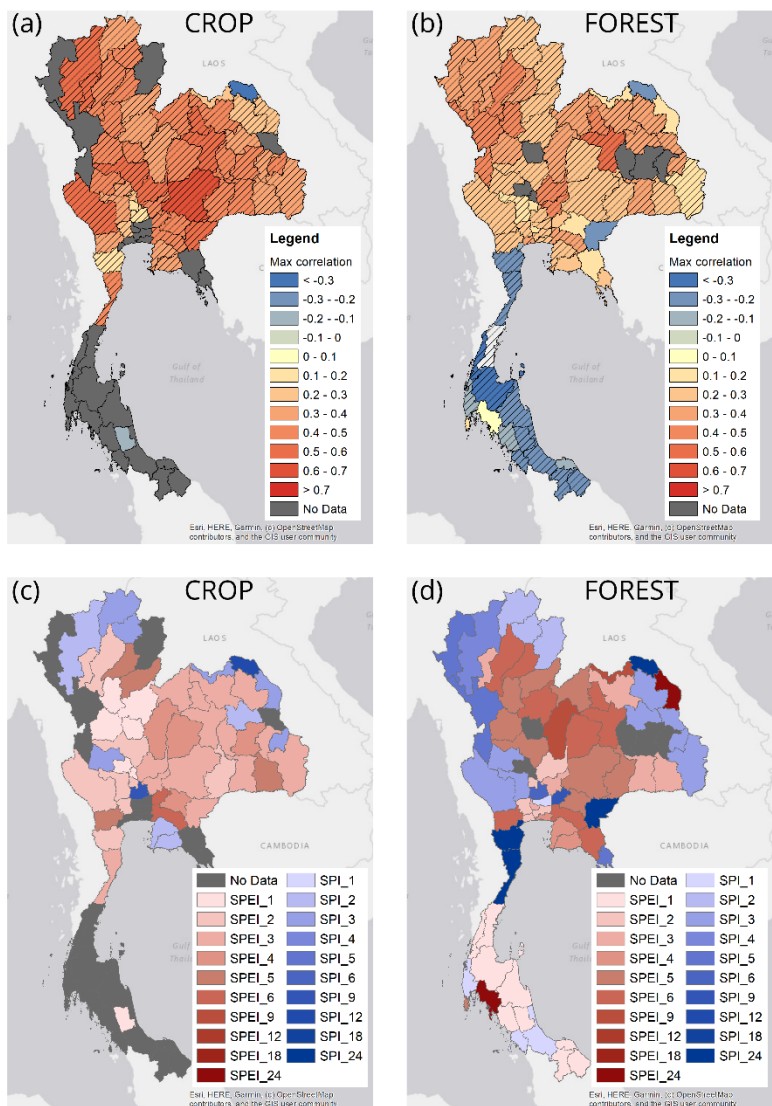

**Figure 5: For the dry season, maximum correlation (all combinations of meteorological indicator with VCI) for each province for (a) crops and (b) forest; and the corresponding meteorological indicator and accumulation period for each province for (c) crops and (d) forest.**

### 3.1.2.2 Wet season

Figure 6 shows the highest correlation for all the combinations of meteorological indicators with VCI for the wet season (Fig.6a and b) with the corresponding meteorological indicator (Fig. 6c and d), for crops (Fig 6a and c) and for forest (Fig 6b and d). The maximum correlation is, in general, lower than for the dry season, which indicates that the impact of meteorological droughts on crops and forest is less severe during the wet season.

A clear difference between crop and forest can be observed. Whereas crops suffer some negative impact from meteorological drought (positive correlations) even during the wet season, forest growth seems to benefit from short droughts (negative correlations in most provinces, with short accumulation period ranking first in many provinces).

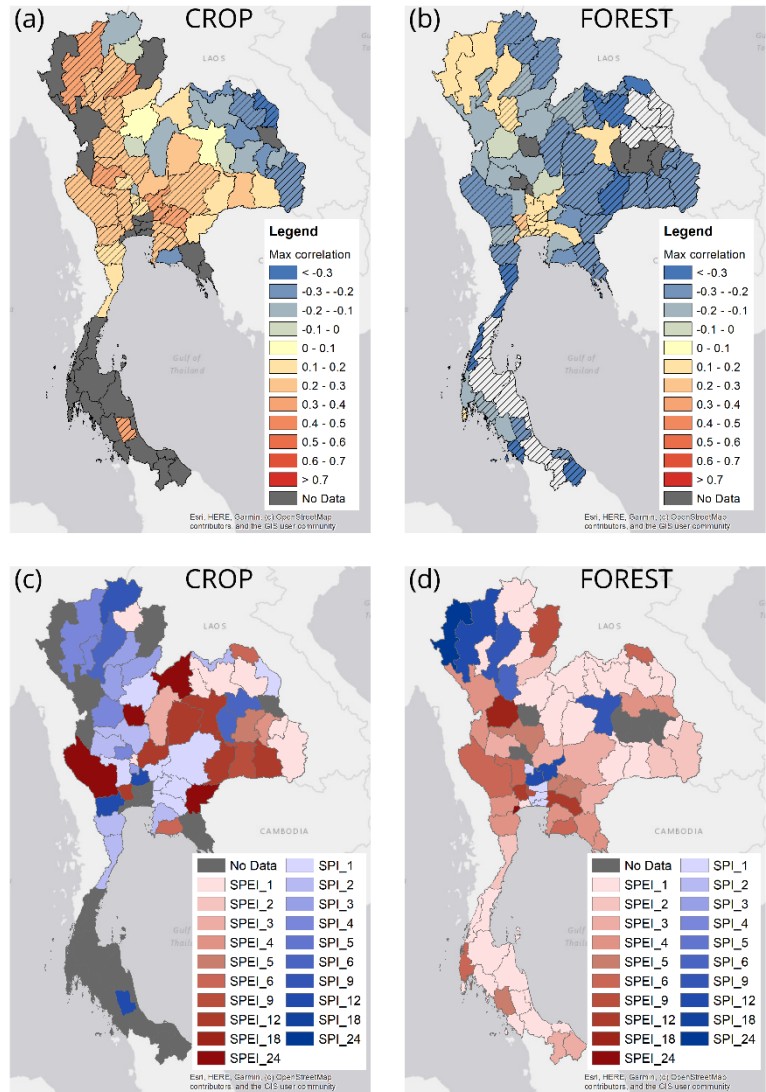

**Figure 6: For the wet season, maximum correlation (all combinations of meteorological indicator with VCI) for each province for (a) crops and (b) forest; and the corresponding meteorological indicator and accumulation period for each province for (c) crops and (d) forest.**

The figures corresponding to Fig.5 and 6 for VHI can be found in the SI (Fig.SF2 and Fig.SF3).

### 3.2 Simulating Crop Productivity

#### 3.2.1 Model performance

The ability for RFs to simulate crop yield in Thailand varied across the country, and between different crops. Cassava productivity was simulated well and RFs were able to explain variance in the data across the country apart from in the S region where the lack of cassava being grown in this region made approximating the relationships between indicators and impacts difficult (Fig. 7a). The variance in cassava yield data explained by the indicators also varied across the country, with more variance explained in the E and NE regions than the W region.

Indicators were also able to explain more than 33% of the observed variance in Corn (S1, S2 and Mixed) yield in the N region, and S region for Corn S2 (Fig. SF4). Furthermore, the amount of variance in paddy rice yield data explained by the RF models only exceeded 33% in the N region (Fig. 7b). In total, variance explained exceeded 33% for five crops in the N region, one crop (cassava) in the NE region, no crops in the W region, one crop (cassava) in the C region, two crops in the E region, and no crops in the S region.

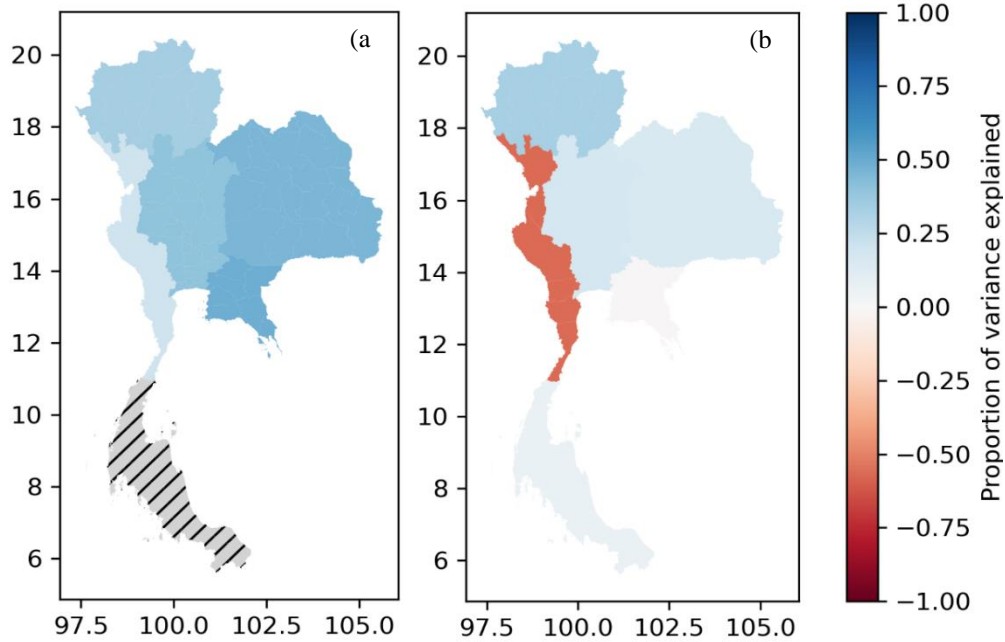

**Figure 7: Amount of variance explained in cassava (a) and paddy rice (b) productivity data by the random forest models.**

#### 3.2.2 Feature importance

RF models also allow us to investigate the relative importance of features (i.e. indicators) in explaining the variance in crop yield by calculating the average decrease in Gini Impurity from the exclusion of individual features, as described in section

2.3.2. To ensure the feature importances were representative of the variation observed within the crop yield data, only RFs that

explained >33% of the variance observed in the crop productivity data were selected for analysis. This resulted in the feature importances of nine RFs being presented here.

Figure 8 shows feature importances in each region for all available crops aggregated, whereas Figure 9 presents the feature importances in each region for a single crop (Cassava), and finally Figure 10 shows feature importances for five distinct crops in region N only. Figure SF4 in the SI also shows the feature importances in two regions for the various types of corn.

Long accumulation periods are assigned relatively high importance in the N region compared to other regions (Fig. 8). This agrees with regional differences observed in cassava feature importances, where SPI24 had the highest mean decrease in impurity (i.e. feature importance) in the cassava model for region N (Fig. 9). Furthermore, 22 SPI indicators (different accumulation periods, and different time of year) were used in the N region RFs, more than any other indicator (11 SPEI, 10 VCI, 6 TCI; Fig. 10). For corn S1, S2 and mixed corn, SPI consisted of 5/9, 5/9 and 4/8 of the indicators used to simulate crop yield, respectively. In contrast, just 3/10 features used in the cassava model were SPI, the remaining features a combination of SPEI, TCI and VCI. Differences in accumulation periods of the most important indicators were also observed between crops. Whereas the corn models (all types) had many long-accumulation period indicators, cassava included eight indicators covering just 1-month, demonstrating the importance of short-term effects on cassava production. Paddy rice also exhibits high importance for several indicators of short accumulation periods. However, in contrast to cassava, 5/10 features selected by the model for paddy rice yield simulation were SPI.

Mean decreases in impurity for cassava models appear to exhibit seasonality in the NE and E regions, with a focus on October-March in NE region, and August-January in the E region (Fig. 9 and Fig. SF4). However, seasonality is less evident in the N and C regions. Seasonality is also observed in the correlation coefficients between cassava yields and meteorological indicators in the NE and E regions (Fig. SF5). Figure SF5 shows that meteorological indicators accumulated between October and April exhibit the highest correlation coefficients in the NE region, agreeing with the results from the RF analysis. In contrast, the highest correlation coefficients occur for indicators calculated from data in February-July in the E region. This period is only covered by a single indicator in the RF model, assigning relatively low importance for estimating cassava yields. This may be due to crop yields being more sensitive to floods in region E, supported by the strong negative correlations between meteorological indicators and cassava yields during August-December (Fig. SF5). This is also consistent with findings from Venkatappa et al. (2021), who found that floods caused more damage than droughts to crops in the Eastern region.

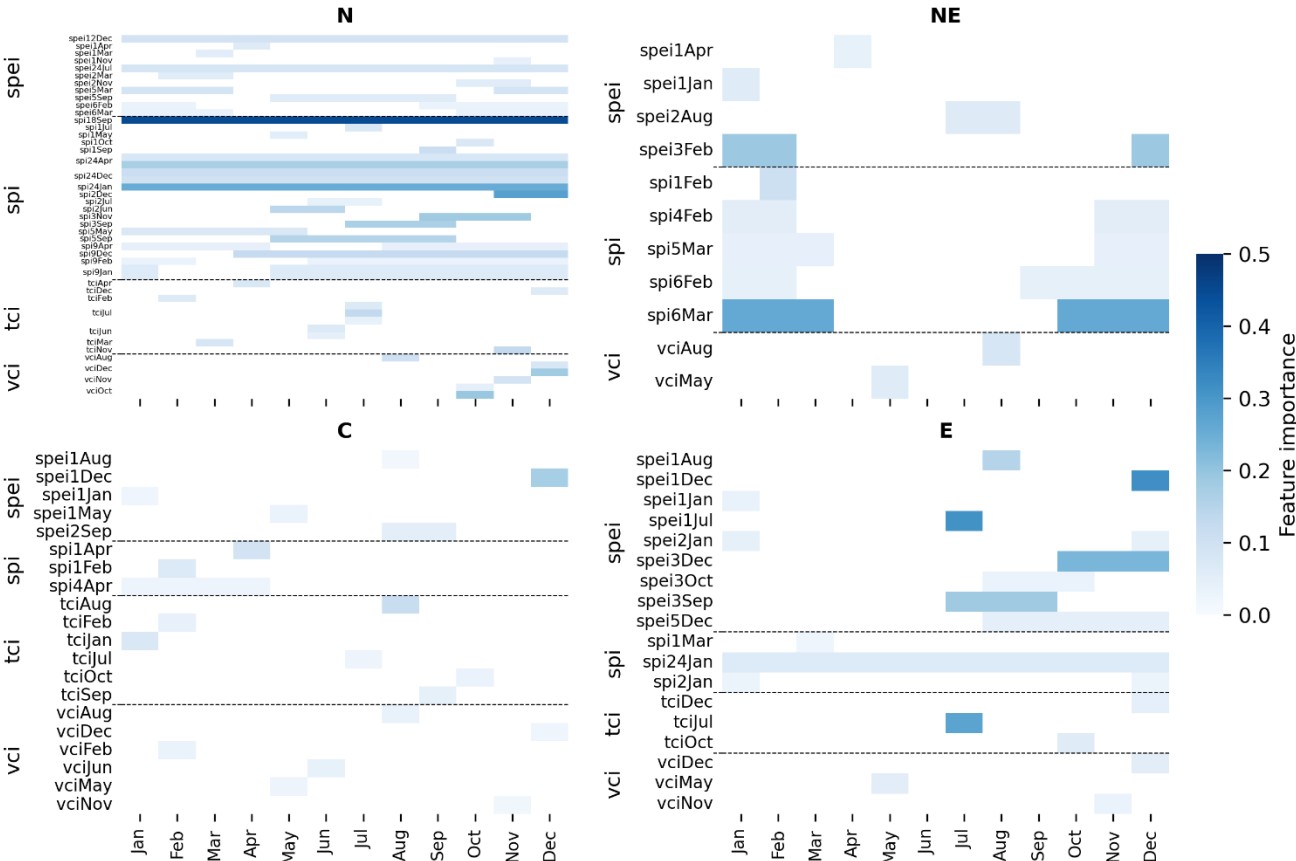

**Figure 8: Heatmap displaying the relative feature importance (impurity decrease) of each indicator used in the random forest models (for all crops) for each region. Each row corresponds to a different indicator, with the y-axis representing the indicator and the length of the bar representing the accumulation period. The x-axis indicates the time of year (month) when the indicator is most relevant for predicting crop yield. For instance, spi6Mar in the NE region represents SPI with a 6-month accumulation period for March, and the bar covers October to March (i.e. the six month period ending in March). The bars are shaded darker for indicators that are more important in the models. Unlike Fig. 9 and 10, which show only one crop per subplot, this figure includes all crops that can be modelled in each region. Region N has five models (cassava, corn S1, corn S2, mixed corn, and paddy rice models), while Region NE, Region C, and Region E have one, one, and two models respectively. The number of rows (i.e., indicators) in each subplot is a consequence of the number of models in each region and the number of variables in each model. The thickness of the lines is a result of the number of indicators displayed for each region and has no meaning attached. Finally, note that different crop models within a region can use the same indicators, leading to some indicators being repeated and having multiple rows within the same region (e.g., vciDec in Region N).**

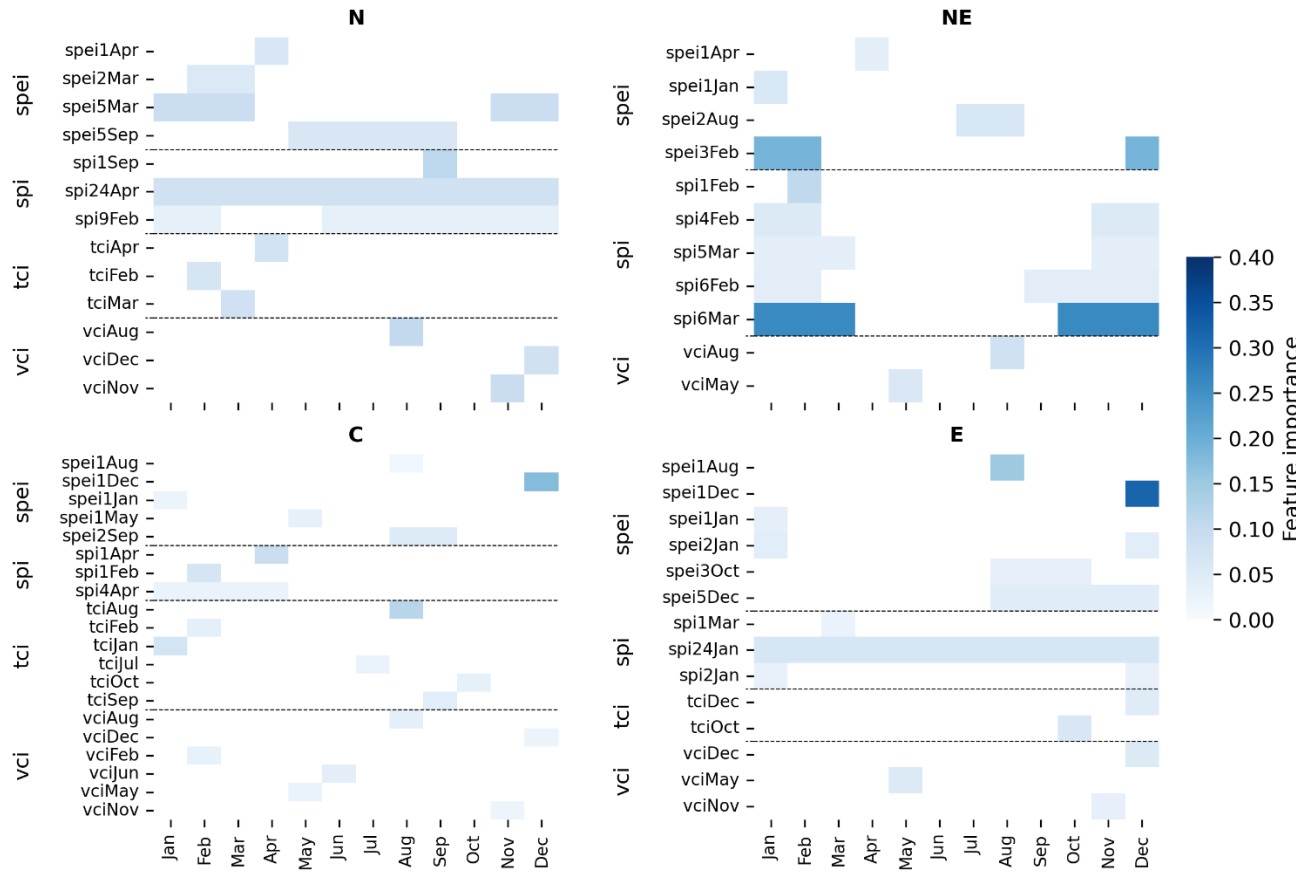

**Figure 9: Heatmap displaying the relative feature importance (impurity decrease) of each indicator used in the random forest models for Cassava for each region. Each row corresponds to a different indicator, with the y-axis representing the indicator and the length of the bar representing the accumulation period. The x-axis indicates the time of year (month) when the indicator is most relevant for predicting crop yield. For instance, spi6Mar in region NE represents SPI with a 6-month accumulation period for March, and the bar covers October to March (i.e. the six month period ending in March). The bars are shaded darker for indicators that are more important in the models. Unlike Figure 8, each subplot here shows only cassava models for each region. However, the number of indicators can still differ between models due to the feature selection process that eliminates highly correlated indicators, which may vary between regions. The number of rows (i.e., indicators) in each subplot reflects the number of variables in each model, and the thickness of the lines is a result of the number of indicators displayed for each region and has no meaning attached.**

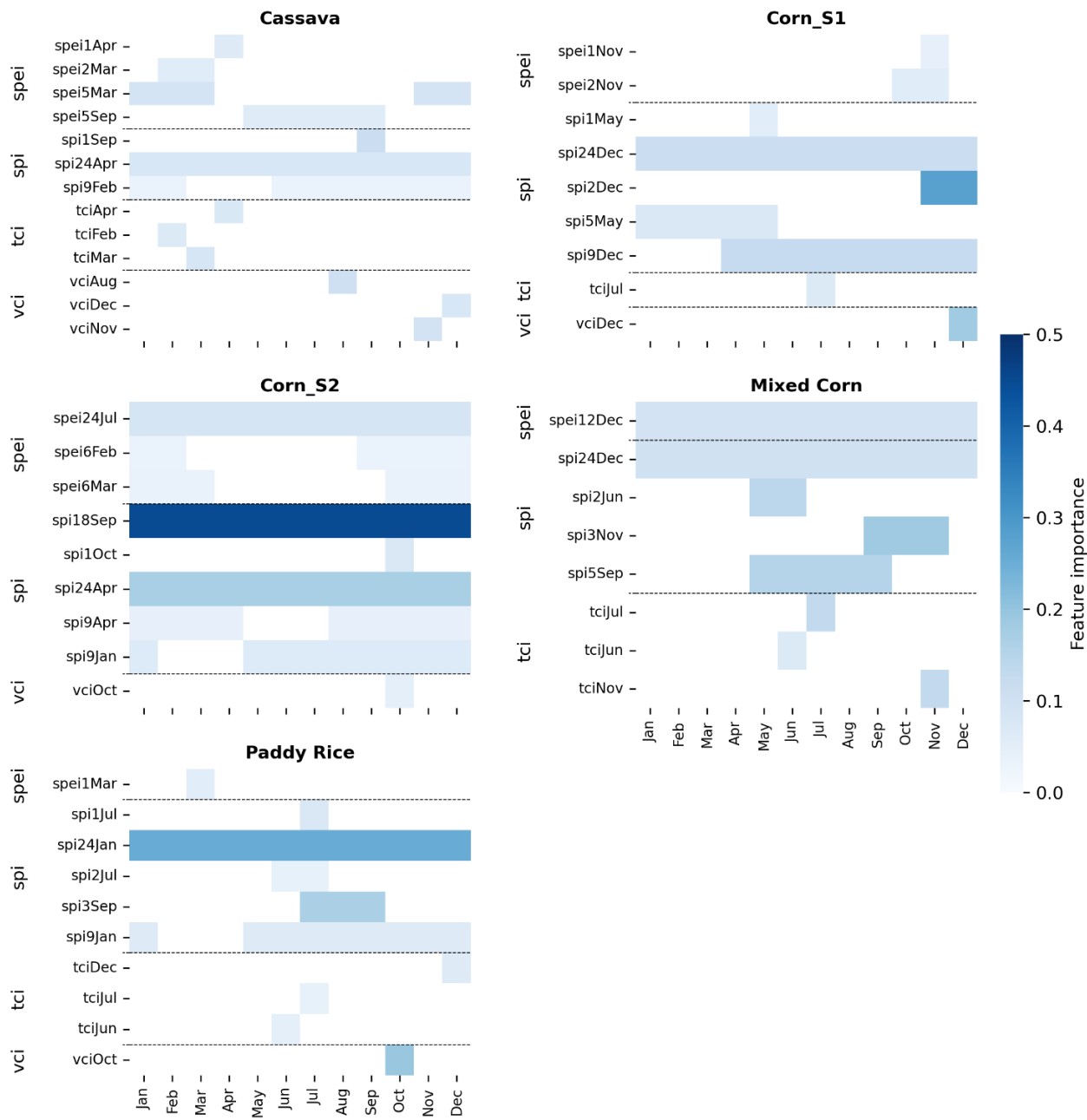

**Figure 10: Heatmap displaying the relative feature importance (impurity decrease) of each indicator used in the random forest models for five different crops in region N. Each row corresponds to a different indicator, with the y-axis representing the indicator and the length of the bar representing the accumulation period. The x-axis indicates the time of year (month) when the indicator is most relevant for predicting crop yield. For instance, spi5Sep for Mixed Corn represents SPI with a 5-month accumulation period for September, and the bar covers May to September ((i.e. the five month period ending in September). The bars are shaded darker for indicators that are more important in the models. Unlike Figure 8, each subplot here only shows the model for a single crop in region N. The number of rows (i.e., indicators) in each subplot reflects the number of variables in each model, and the thickness of the lines is a result of the number of indicators displayed and has no meaning attached.**

## 4. Discussion

### 4.1 Spatial variation in indicator-to-impact relationships

The correlation analysis showed that, in most of the provinces in region N (Fig.5c and 6c), SPI is more correlated with crop yield than SPEI, whereas SPEI dominates in the rest of the country during both the wet and dry seasons. This, combined with the high importance placed on SPI features by the RF models in region N (Fig.8 and 9), demonstrates the strong relationship between SPI and crop yield, particularly compared to other parts of the country. Region N is one of the most irrigated areas of Thailand together with region C (FAO, 1999; Varawoot, 2016). A region's dependence on irrigation (and therefore water storage) seems to result in (i) a lack of variation in indicator importance across different seasons; (ii) an importance of long accumulation periods (when storage gets depleted); and (iii) SPI being more important than SPEI to explain drought impacts. This last point could be explained by the fact that low precipitation in region N leads to the Actual Evapotranspiration (AET) to be water limited (i.e. AET < PET) meaning SPEI may be less closely linked to agrometeorological conditions. However, this is also the case for region W and C, and to lesser extend NE as well, where precipitation is also low, but in these other regions, SPEI's importance is generally dominant (Fig.5c). Therefore, it is likely that the dominant importance of SPI in region N is linked to the reliance of water storage for irrigation in this region, particularly for Corn S2 which is planted in the dry season and relies heavily on irrigation (Fig.10). Therefore, a deficit in rainfall (and consequent depleted storage) will have a strong impact on crop yield. For regions where rainfed crops dominate, shorter droughts can cause impacts and SPEI becomes a stronger explanatory variable, given the effect of the increased evaporative demand. This was observed in regions E and NE, where SPEI indicators had high importance; whereas in regions N and C, SPI and TCI were more important for understanding impacts of droughts on cassava yield. These results agree with previous studies which found strong relationships between SPEI and crop yield in the Mun River Basin located in the NE region (Khadka et al., 2021; Prabnakorn et al., 2018).

Region N was the only region where we successfully build RF models for five crops (Cassava, Corn S1, Corn S2, Mixed Corn, Paddy rice). This was due to a combination of a lack and/or low quality of yield data, limited crop area, relationships being too complicated (with other factors than drought affecting crop yield), or indicators not being related to impacts in other regions. The comparison of the feature importance for these five models in N region provides insight on the differences between crops. Both short- and long duration droughts are important for all the crops simulated, demonstrated by the presence of indicators with 1-12 month accumulation periods for each crop. However, there was a higher prominence of short-accumulation period indicators for cassava than other crops, particularly corn (all types). Cassava has the longest crop calendar (12 months) therefore having a higher chance of experiencing drought during the growing period. Other crops have shorter cycle compared to Cassava (4-5.5 months for paddy rice, 4 months for both corn S1 and S2). Also, Cassava is the least water-demanding crop of the list (irrigation requirement of around 20m3/ton in the wet season and 65m3/ton in the dry season, Gheewala et al., 2014). This explains the comparatively lower importance of long accumulation indicators for Cassava (Fig.10), given less reliance on water storage, especially compared with the most water-intensive crops, such as paddy rice (irrigation requirement of 520m3/ton during the wet season and 1140m3/ton in the dry season) and Corn S2 (irrigation

requirement of 850m3/ton in the dry season). Whereas SPI is important for simulating productivity of corn and paddy rice in the N region; SPEI, TCI and VCI were more important in simulating cassava productivity than in other crops. These differences in results between regions and crops demonstrates the importance of having region and crop-specific policies and actions and indicators for drought monitoring.

## 4.2 Temporal and crop specific variation in indicator-to-impact relationships

When we look at seasonal differences, we observe that SPI has higher correlation with yield during the wet season, whereas SPEI is generally more correlated during the dry season (Fig. 5c and 6c). This suggests the importance to account for both temperature and evapotranspiration in dry season monitoring. We also observe that the highest correlations are for longer accumulation periods in the wet season compared to the dry season. This might be because longer wet season accumulation periods also include the information of the preceding dry season. The N region paddy rice RF model exhibited high importance in indicators accumulated for the second half of the year (Fig. 10). Rice can be harvested twice or even three times per year in certain regions. The main rice crop cycle has its growing season between June and December, which has a large part during the wet season. The RF model for paddy rice in region N exhibits high importance during this period (Fig. 10).

The RF models show spatial variations in the strength of relationships between crop yields and different accumulation periods, with longer accumulations assigned higher importance in region N, compared to other regions (Fig. 8 and 9), and is also evident in the correlation analysis plot (Fig. SF5). This might be partly explained by the presence of major dams in that region which can mitigate the effect of short droughts (LePoer, 1987). The comparison of feature importance for cassava and paddy in region N (Fig. 9) supports the idea that different indicators and periods are important for each crop.

Cassava models for regions NE and E exhibit importance during October-March (Fig. 9), which overlaps with the dry season. Cassava is usually planted in April-June, and its yield is known to be sensitive to water stress in early stages of growth, corresponding to root initiation and bulking (Connor et al., 1981; Okogbenin et al., 2013; Oliveira et al., 1982). This seasonal pattern is not seen as clearly for regions N and C, and is most probably explained by the fact that these two regions rely heavily on irrigation as opposed to regions NE and E which are mainly rainfed.

Paddy rice's critical period for drought stress – which will have a severe effect on crop yield – are the early stages of germination and seedling stage, and also the flowering period (Farooq et al., 2012; Kadam et al., 2017; Mishra & Panda, 2017; Yang et al., 2019). However, it should be noted that this effect varies significantly depending on the specific crop variety, and the increasing adoption of drought resistant varieties mitigates the impacts. The RF model for paddy rice in region N (Fig. 10) shows important features overlapping with these critical periods, in particular the early stage of the germination and seedling.

The critical period for drought for corn (all types) is in the early period of the growing season, with water stress after the anthesis (flowering) having no significant impact on crop yield (Pradawet et al., 2023). However, no distinct seasonality is observed in feature importance for corn in region N (Fig. SF4), with a high prominence of long accumulation periods, despite corn being grown within mostly one season. This most likely reflects the dependence to irrigation of this crop in region N.

However, in region E which is dominated by rainfed crops, feature importance is concentrated around this critical period (June-

500 October, Fig. SF4).

We were not able to build RF models able to simulate longan productivity. The lack of model skill could be due to several factors. Firstly, unlike the other crops studied here, longan is a tree, and the effect of drought might be more complex. Secondly, longan is not grown extensively in comparison to the other crops, and the resulting lack of data might make it hard to identify patterns. And finally, longan trees are particularly sensitive to drought during the flowering and early fruit development stages

(Menzel & Waite, 2005), which coincides with the dry season in Thailand, making this crop completely reliant on irrigation for production (Spreer et al., 2013). Irrigation mitigates the effects of drought, making it more difficult to model direct effects of meteorological droughts on crop production.

For the crops where it was possible to build a RF model, the analysis of the temporal variation in feature importance and the indicator-to-impact relationships provide insights into critical periods of the year for early warning of impacts and relevant

accumulation period. Specifically, these are periods of interest when dry conditions could lead to impacts.

## 4.3 Crop vs. forest

Clear differences on how drought impacts crops versus natural vegetation (forest) were observed. Firstly, a geographic distinction between the North (regions N, NE, W, C and E) and the South (region S), and secondly, temporal differences between wet and dry seasons, suggest that drought management will be different between regions and time of year. The

515 summary of these differences can be found in Table 2, which also indicates the most relevant drought indicator to be used in a DMEW context for each land use type, season and region.

For forests in the North, droughts during the dry season will limit vegetation growth. This is in line with previous studies on dendochronology which have found that moisture availability in pre-monsoon season is the predominant climatic factor controlling vegetation growth of tree species in tropical Southeast Asia, though there are important differences between species

(e.g. Buckley et al., 1995; Rakthai et al., 2020; Sano et al., 2008). Generally, the indicator showing the highest correlation with impacts is for longer accumulation period for forests than for crops, suggesting that shorter droughts will have impacts on crops whereas only longer droughts will affect forests. The higher resilience to droughts of forests compared to crops is at least partially explained by the deeper root systems of forest trees allowing them to extract water from deeper layers of the soil (Bréda et al., 2006; Schenk & Jackson, 2002).

During the wet season in the North, and year round in the South, forests do not suffer from drought (negative correlation), and short droughts might even contribute positively to vegetation growth. Roebroek et al. (2020) produced a global distribution of hydrologic controls on forest growth. For Thailand, the forest growth in the South is mainly energy-limited (solar radiation), rather than water-limited. Hence short droughts, which are associated with increased radiation (due to decreased cloud cover) can have a beneficial effect to the forest.

The North of Thailand is a mixture of water-limited areas (water is the limiting factor for vegetation growth) – which explains the negative effect of droughts during the dry season – and oxygen limited areas (when growth is limited by the availability of

oxygen by the root, often due to flooding), which in Thailand is typical during the wet season. This explains the positive effect that short droughts have on forests during the wet season in the North (less flooding).

For crops, droughts have a negative impact both in the dry and wet seasons, though the effects during the dry season are stronger. The correlation could only be derived in the North, as the South is dominated by forested areas. Although SPEI shows the highest correlation to crop yields in most cases in the North, SPI is more prominent in region N. This, combined with the relatively high correlation between SPI and crop yields in region N during the wet season (Fig. SF5), explains the higher importance placed on SPI compared to SPEI in the RF results. This can be explained by the higher dependency on storage/irrigation in region N, than in the rest of the country.

**Table 2: Summary of main findings on relationship between drought indicators and drought impacts, and differences between land cover, region and season.**

| | CROP | | FOREST | |
|---|---|---|---|---|
| | North (regions N, NE, W, C and E) | South (region S) | North | South |
| Dry season | Strong negative impact of droughts.<br><br>Most correlated indicator: mostly SPEI, except for region N where SPI is more prominent; short to medium accumulation period. | No data (Crop area too limited) | Negative impact of droughts.<br><br>Most correlated indicator: mostly SPEI (some SPI), medium to long accumulation periods | Positive impact of drought.<br><br>Most correlated indicator: SPEI, short accumulation period. |
| Wet season | Negative impact of droughts, though less strong than in dry season.<br><br>Most correlated indicator: combination of SPI and SPEI, different length of accumulation period | No data (Crop area too limited) | Positive impact of drought in most of the North. Negative impact of drought in NW.<br><br>Most correlated indicator: mostly SPEI, short accumulation periods (except NW: SPI long accumulation period) | Positive impact of drought.<br><br>Most correlated indicator: SPEI, short accumulation periods. |

## 4.4 Limitations

Though this study provides important new insight into the relationship between drought indicators and drought impacts in Thailand, some limitations should be acknowledged.

Firstly, there are limitations due to the imperfect nature of the data. By averaging VIs at province level, and correlating these values to crop yield from the dominant crop within that region, we inevitably introduce some noise, especially in provinces with a varied range of cultivated crops. However, a detailed map of crop distribution was not available. Therefore, the simplified approach taken here of using a land cover mask differentiating cropland from forests (but with no distinction between crops) was the best possible with the available data.

In addition, the time series available to carry out our analysis are short (15 years), which makes them more susceptible to noise, and also means that fewer drought events are available to learn from. Nevertheless, the study period (1984-2019) does include some of the major recent drought events such as 1990-1993, 1997, 2005, 2008 and 2015-2016.

Secondly, some limitations come from the methods we have used. Different factors can cause a trend (e.g. climate change, policy change, improvement in agricultural practice, etc.). However, we have applied a simple detrending approach (linear regression) which assumes that the trend is linear. In addition, the short length of the record makes it difficult to identify any trend. Another methodological limitation is our use of correlations to link drought indicators to drought impacts. Correlations can only explain linear relationships. However, the reality can be more complex, especially when looking at precipitation, where both extremes (droughts and floods) can have similar effects on crop yield loss. RFs are powerful tools for producing predictive models from data, but they are considered 'black boxes' since they do not explicitly extract the relationships between input features and the predicted outcomes. However, RFs can aid in the interpretation of the model through the analysis of feature importance, which identifies the most influential variables in making predictions. In addition, we only consider drought indicators and VIs as input variables, but many other factors can influence crop yield, such as floods, low temperature, disease, policy changes, farming practices, etc. This might also partially explain the relatively low performance of some of the RF models. Lastly, another limitation of using data-driven models such as RFs is the need for a large amount of data needed to train the model effectively. In our study, we had a relatively short period of data available, which limited the amount of data available for training the models. As a result, the models may not have been able to accurately capture the full range of conditions that could occur in the real world. For example, for species such as longan, which are more susceptible to long drought events, the limited instances of these events in our training data may have affected the model's ability to accurately predict impacts.

## 4.5 Future work

In this study, we used RF models primarily to analyse the relationships between drought indicators and impacts, and to identify the relative importance and timing of relevant indicators for impacts on crops and forests. While the main focus of our analysis was on feature importance, our analysis also demonstrated the potential of RFs to simulate unseen data, which suggests they could be used for impact prediction. With further work, such as addressing the limitations discussed above, these models could be used for DMEWSs, support and compensation schemes, long-term planning, etc.

Furthermore, alternative methods could be explored and compared with the ones used here. Simpler approaches could provide simpler interpretation, such as the logistic regression (model diagnosis and equifinality/extrapolation). Due to the linear additive nature of logistic regression, it can be used to identify thresholds at which drought impacts are expected (Bachmair et al., 2016b; Parsons et al., 2019). However, for that same reason, it can only account for the probability of impacts increasing as conditions get drier or wetter, not both. Given that Thailand suffers from both floods and droughts, more complex models capable of capturing this non-linearity would be more suitable.

The RF models developed here offer promising results, but could be compared to more sophisticated approaches. The use of machine learning/deep learning algorithms (Artificial Intelligence, AI) and Bayesian inference techniques are currently two rapidly developing areas of research, and are increasingly used in environmental science. AI is very effective in finding patterns and connections within large volumes of multi-source spatio-temporal information, while Bayesian models are well suited for modelling complex spatio-temporal variations and capturing uncertainties. Shen et al. (2019) used Deep Learning technique

(Artificial Neural Network) to build a drought monitoring model in China, whereas Bouras et al. (2021) developed a crop yield forecasting tool based on eXtreme Gradient Boost (XGBoost) in Morocco. Salakpi et al. (2022) on the other hand, used a dynamic hierarchical Bayesian approach for forecasting vegetation condition in Kenya. With the burgeoning of new and increasingly complex methods, an assessment of the most suited approach in the context of DMEWS would be highly valuable.

## 5. Conclusions

In this study, we used a combination of traditional statistical approaches and machine learning techniques to analyse the relationship between drought indicators and drought impacts on vegetation and crops. These approaches are relatively novel in environmental science, particularly in south-east Asia context, bridging the gap between hazard and vulnerability by incorporating observed drought impact data.

Firstly, we carried out a correlation analysis to study the link between meteorological drought indicators (SPI and SPEI at

different accumulation periods) and remote sensing vegetation indices (VCI and VHI) used as proxy of crop yield and forest growth. Our analysis shows that these links in Thailand vary greatly depending on the land use (crops vs. forest), season (wet vs. dry) and geographical region, as does the type of droughts (short vs. long duration, with or without high temperature) which causes most damaging impacts. Some of the main findings are that droughts have a negative effect on crops both during the wet and dry seasons, though the length of the droughts having most impact differ between seasons (shorter droughts during

the dry season). Results also highlighted that short droughts can have a beneficial effect on forest growth in the wettest areas of the country and during the wet season.

Secondly, we built a series of random forest models to estimate crop productivity for each crop and region separately. This allowed a more in-depth analysis of the importance of the different drought indicators in a crop-specific way. The analysis of feature importance has teased out seasonal patterns of feature importance for individual crops, often linked to their growing

season, though the presence of irrigation systems in some of the regions (regions N and C) remove some of that seasonality. This new knowledge about the importance of specific drought indicators to predict drought impacts for targeted crops and regions could be used to improve drought monitoring and early warning systems in Thailand, particularly for the agricultural sector which is both economically important to Thailand as well as vulnerable to the drought, as it will allow tailored monitoring of the most relevant indicators for individual crop/region. The best indicators to monitor vary in space and time,

as well as by land use and crop type. The work presented in this paper can provide guidance and inform water managers on the best indicator to use spatio-temporally, which ultimately will contribute to increase Thailand's resilience and preparedness

to droughts. Furthermore, the methodology can be replicated in other areas of the world to help build this knowledge in other countries aiming to increase their resilience to droughts.

## Acknowledgment

This project was funded through a Cranfield University Global Challenges Research Fund QR 2020/21 grant and a joint grant from NERC (Natural Environment Research Council, UK), grant number NE/S003223/1 and TSRI (Thailand Science Research and Innovation), grant number RDG6130017 for the STAR project (Strengthening Thailand's Agricultural drought Resilience). The funding was also complemented by UKCEH's Land Air Water International Science (LAWIS), and NC-international programmes [NE/X006247/1] delivering National Capability, both funded by NERC. The authors would also

like to express their gratitude to the two reviewers, Samuel Jonson Sutanto and Veit Blauhut, for their valuable feedback which has contributed to improve the quality of the manuscript.

## Code/Data availability

All code and data are available upon request. Open data used in this study is listed in Table 1.

## Author contributions

**Maliko Tanguy:** Conceptualization, Formal Analysis, Investigation, Methodology, Supervision, Validation, Visualization, Roles/Writing – original draft, Writing – review & editing

**Michael Eastman:** Conceptualization, Formal Analysis, Investigation, Methodology, Validation, Visualization, Roles/Writing – original draft

**Eugene Magee:** Data Curation, Formal Analysis, Investigation, Methodology, Validation, Visualization, Roles/Writing –

original draft

**Lucy Barker:** Conceptualization, Writing – review & editing

**Thomas Chitson:** Data Curation

**Chaiwat Ekkawatpanit:** Funding acquisition, Writing – review & editing

**Daniel Goodwin:** Conceptualization, Methodology, Writing – review & editing

**Jamie Hannaford:** Conceptualization, Funding acquisition, Supervision, Writing – review & editing

**Ian Holman:** Conceptualization, Funding acquisition, Writing – review & editing

**Liwa Pardthaisong:** Funding acquisition, Writing – review & editing

**Simon Parry:** Conceptualization, Funding acquisition, Project administration, Resources, Writing – review & editing

**Dolores Rey Vicario:** Funding acquisition, Writing – review & editing

**Supattra Visessri:** Funding acquisition, Project administration, Writing – review & editing

**Competing interests**

The authors declare that they have no known competing financial interests or personal relationships that could have appeared to influence the work reported in this paper.

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
