# Peer review of "Indicator-to-impact links to help improve agricultural drought preparedness in Thailand"

_EGUsphere, 2023_

## Author Response (AR1)

**RESPONSE TO REVIEWER #1 (Samuel Jonson Sutanto)**

We would like to express our gratitude to Samuel for taking the time to review our paper and for providing valuable feedback and suggestions. We appreciate the thoroughness of your review, which has significantly contributed to improving the quality of the manuscript.

Below is a point-by-point response to all comments. Original comments are in **black**, whereas the authors' responses are in **blue**, and changes made in the manuscript are in **red**.

**Title:** Indicator-to-impact links to help improve agricultural drought preparedness in Thailand

**Authors:** Tanguy et al.

**Recommendation:** minor revision

**Summary**

This paper correlates meteorological drought indices, represented by SPI and SPEI, and vegetation indices such as VCI, TCI, and VHI with forest growth and crop yield impacts. Two approaches were used in the analysis, which are the Pearson correlation and the Random Forest machine learning model. The authors found that the strength of correlations depends on land use, season, region, and drought duration. Crops are strongly impacted by drought in both wet and dry seasons. The impact of droughts, however, is less apparent for forest growth. The use of the Random Forest technique allows a more in-depth analysis of the importance of different drought and vegetation indicators. The authors also highlighted that the knowledge of linking specific indicators to the drought's impact on crops will help to improve the DMEWS and perform mitigation actions.

**Assessment**

This paper analyzes the use of different drought and vegetation indicators to link these indices with the impact of drought on crop yields and forest growth. The manuscript is interesting and well written. I have a few minor comments below and two general comments, but only for clarification. I believe this work is well suited for NHESS.

Thank you for the encouraging and positive feedback.

**General Comments**

I have two general comments regarding the manuscript but all of them are only for clarification and improvement of the manuscript.

1. I am wondering why the authors used the Random Forest (RF) approach only to find the importance of all indicators on crop yields and forest growth. RF can also be used to predict the crop yields by: 1) training the predictor variables, here are e.g., drought and vegetation indices, and the response variables (e.g., crop yields), resulting in crop yield impact model; 2) using the developed model to predict the impact of drought on crop yield by leaving out the predicted year from the training period. Maybe it is interesting to do this since the authors already have the script to develop the RF model and mentioned this in Figure 2. The authors can train the RF model again without the predicted year and in the end forecast the yields and validate the result with the observed crop yield data. Otherwise, it is worth to discuss the use of machine learning to predict crop yield and not only to find the importance of predictor variables.

Thank you for your comment. We agree that RFs can be used to predict crop yield and we have indeed used it in that way, as mentioned in line 221, where we state that "*Regional Random Forest (RF) models were used to predict agricultural impacts (crop yield).*" However, we appreciate the suggestion to explore this application of RFs more explicitly in our manuscript. To make it clearer that the RFs were used to predict crop yield, although this was not the main focus of our study, we have added the following sentence to the end of section 2.3.2:

"*While RFs were built to predict crop yields, the main focus of our study is their use to study feature importance to identify monitoring priorities for different regions and crops.*"

Regarding the suggestion to use leave-one-out cross-validation, we used 5-fold cross-validation instead, as we explained in lines 243-245. Our choice was based on computational efficiency, as 5-fold cross-validation is faster than leave-one-out cross-validation.

We also agree that we could have emphasised more the potential of RFs to predict drought impacts (crop yields in our case). To address this, we have rephrased the first paragraph of the "Future Work" section (line 511-514) as follows:

"*In this study, we used RF models primarily to analyse the relationships between drought indicators and impacts, and to identify the relative importance and timing of relevant indicators for impacts on crops and forests. While the main focus of our analysis was on feature importance, our analysis also demonstrated the potential of RFs to simulate unseen data, which suggests they could be used for impact prediction. With further work, such as addressing the limitations discussed above, these models*

*could be used for DMEWSs, support and compensation schemes, long-term planning, etc."*

2. I have difficulty to understand figures 8, 9, and 10. I read the caption over and over again but still cannot interpret the figures. Is there any other way to present your results in a simple manner, so thus the readers can understand the results? For example, it is not clear to me why some lines are thick, and some are thin. Also, why VCI N has 6 thin lines and VCI E has only 3 thick lines? How to indicate 24 months accumulation periods in the results? Maybe modify the Y-axis?

Thank you for your valuable feedback regarding the difficulty in interpreting figures 8, 9, and 10. We acknowledge the deficiencies in these plots and have carefully considered the best way to present our results. We have developed two alternatives, one that is a slightly modified version of the existing plots and a second that is an alternative version of the heatmaps. We will present both here, but we have included the modified version of the existing plots for the revised version. We have also provided a more detailed explanation in the caption to improve the interpretation of the figures.

Regarding the number of lines in the plots, we apologize for the lack of clarity in our original caption. Figure 8 shows the feature importance for all variables used in all models within each region. The number of variables per model and the number of models vary across regions and crops, resulting in different numbers of lines for each subplot. Specifically, the number of crops modelled per region varies, and RFs were not able to model some of the crops in some of the regions. For example, in the N region, we modelled 5 crops, but in the NE region, we were only able to model one crop (cassava). The caption has been revised to provide a clearer explanation.

The thickness of the lines does not have any meaning attached to it and only reflects the number of variables in that subplot. There are more variables used in all the models in the N region than in the other regions, resulting in thinner lines in the N subplot. An option would have been to have different sized subplots to have lines with equal thickness. However, this solution would have been much more complicated technically, as we have used python module *seaborn*'s 'heatmap' functionality, which doesn't have that option built in.

We also understand that it was not clear why VCI (for example) has multiple lines in the plots. We separated the VCI for each individual month separately (see line 225), and sometimes VCI for the same month is repeated because it was used in different models for different crops. Hopefully the revised caption explains this better.

Finally, we acknowledge that the accumulation period of 12, 18, and 24 months could not be differentiated in the original plots. In the revised version, we have

added the full variable name explicitly, so the accumulation period can now be identified.

Thank you again for your feedback, which has helped us improve the clarity and interpretation of our results.

**Alternative Figure 8:**

Option 1: We have revised Figure 8 by explicitly labelling each indicator to improve its readability (y-axis). This is the only difference between the revised and original versions. We acknowledge that the labels for Region N may appear small, but we hope that readers can still discern them by zooming in. We believe this is preferable to omitting the labels altogether, as was done in the original version. Additionally, we have updated the caption with more detailed explanations to clarify the information presented in the figure.

[Figure]

**Figure 8: Heatmap displaying the relative feature importance (impurity decrease) of each indicator used in the random forest models (for all crops) for each region. Each row corresponds to a different indicator, with the y-axis representing the indicator and the length of the bar representing the accumulation period. The x-axis indicates the time of year (month) when the indicator is most relevant for predicting crop yield. For instance, spi6Mar in the NE region represents SPI with a 6-month accumulation period for March, and the bar covers October to March (i.e. the six month period ending in March). The bars are shaded darker for indicators that are more important in the models. Unlike Fig. 9 and 10, which show only one crop per subplot, this figure includes all crops that can be modelled in each region. Region N has five models (cassava, corn S1, corn S2, mixed corn, and paddy rice models), while Region NE, Region C, and Region E have one, one, and two models respectively. The number of rows (i.e., indicators) in each subplot is a consequence of the number of models in each region and the number of variables in each model. The thickness of the lines is a result of the number of indicators displayed for each region and has no meaning attached. Finally, note that different crop models within a region can use the same indicators, leading to some indicators being repeated and having multiple rows within the same region (e.g., vciDec in Region N).**

Option 2: We attempted an alternative method of displaying the information in our plots. In these visualisations, we included all indicators, including those that were not used to construct the random forest models (which are greyed out). This approach ensures that each subplot has the same number of indicators and avoids the issue of varying line thicknesses. However, this method omits an important dimension: the period covered by the accumulation period. We rely on this information extensively in our text to explain differences between regions and crops. Consequently, we have decided to retain option 1 for our plots.

[Figure]

**Figure 8: Heatmap depicting the importance of each feature (indicator) for all models in each region. The x-axis shows the indicators, with the accumulation period included where relevant, and the y-axis shows the feature importance for each month of the year. Greyed areas represent indicators that were not used to build any of the models in that region. Unlike Fig. 9 and 10, where only one crop is shown per subplot, each subplot in this figure includes all crops that could be modelled in that region. Region N has five models (cassava, corn S1, corn S2, mixed corn, and paddy rice models), while Region NE, Region C, and Region E have one, one, and two models, respectively.**

**Alternative Figure 9:**

Option 1: Similar to the new Figure 8 (option 1), the only difference between this figure with the original version is the explicit labelling of each indicator, and improved caption.

[Figure]

**Figure 9: Heatmap displaying the relative feature importance (impurity decrease) of each indicator used in the random forest models for Cassava for each region. Each row corresponds to a different indicator, with the y-axis representing the indicator and the length of the bar representing the accumulation period. The x-axis indicates the time of year (month) when the indicator is most relevant for predicting crop yield. For instance, spi6Mar in region NE represents SPI with a 6-month accumulation period for March, and the bar covers October to March (i.e. the six month period ending in March). The bars are shaded darker for indicators that are more important in the models. Unlike Figure 8, each subplot here shows only cassava models for each region. However, the number of indicators can still differ between models due to the feature selection process that eliminates highly correlated indicators, which may vary between regions. The number of rows (i.e., indicators) in each subplot reflects the number of variables in each model, and the thickness of the lines is a result of the number of indicators displayed for each region and has no meaning attached.**

Option 2: Similar to the alternative option 2 for Figure 8, this heatmap has the advantage of displaying an equal number of indicators in each subplot, which can facilitate the comparison of feature importance between regions. However, this alternative visualisation does not include information on the accumulated period of the indicators, which is a relevant aspect for our analysis and discussion. Therefore, we chose to use the improved version of our original heatmap (option 1) in our study, as it provides a more comprehensive representation of the feature importance for each indicator, including its temporal relevance for predicting cassava yield.

[Figure]

**Figure 9: Heatmap depicting the importance of each feature (indicator) for cassava models in each region. The x-axis shows the indicators, with the accumulation period included where relevant, and the y-axis shows the feature importance for each month of the year. Greyed areas represent indicators that were not used to build the cassava model in that region. Note that despite modelling the same crop in each region, the number of indicators can differ between models due to the feature selection process that eliminates highly correlated indicators, which may vary between regions.**

**Alternative Figure 10:**

Option 1: Similar to the new Figure 8 (option 1), the only difference between this figure with the original version is the explicit labelling of each indicator, and improved caption.

[Figure]

**Figure 10: Heatmap displaying the relative feature importance (impurity decrease) of each indicator used in the random forest models for five different crops in region N. Each row corresponds to a different indicator, with the y-axis representing the indicator and the length of the bar representing the accumulation period. The x-axis indicates the time of year (month) when the indicator is most relevant for predicting crop yield. For instance, spi5Sep for Mixed Corn represents SPI with a 5-month accumulation period for September, and the bar covers May to September ((i.e. the five month period ending in September). The bars are shaded darker for indicators that are more important in the models. Unlike Figure 8, each subplot here only shows the model for a single crop in region N. The number of rows (i.e., indicators) in each subplot reflects the number of variables in each model, and the thickness of the lines is a result of the number of indicators displayed and has no meaning attached.**

Option 2: Same comment as Fig. 9, option 2.

[Figure]

**Figure 10: Heatmap depicting the importance of each feature (indicator) for RF models for five different crops in region N. The x-axis shows the indicators, with the accumulation period included where relevant, and the y-axis shows the feature importance for each month of the year. Greyed areas represent indicators that were not used to build the cassava model in that region.**

**Line by line comments**

L refers to line and P refers to page.

**P1L19**: Maybe re-write "…it provides stakeholders…" as "…provided to stakeholders…"?

Thank you. This has now been corrected.

**P2L33**: The authors may add a study on extreme high and low flow events in Southeast Asia including Thailand due to climate change (Hariadi et al., 2023).

Thank you for your suggestion. We agree that this study would be valuable in setting the context for our research by highlighting the expected impact of climate change on high and low flows in the region. We have incorporate the reference you provided into our revised manuscript. Line 34: "*This trend is expected to intensify further in the near future in South-East Asia as highlighted by Hariadi et al. (2023).*"

**P2L46**: Full stop after the ICID reference.

Thank you, this has now been corrected.

**P2L53**: "has" -> "have"

Thank you for spotting this mistake, we have corrected this.

**P3L75**: Double reference from Stahl et al., 2016.

Thank you for pointing this out. We apologise for the confusion. The reference in question actually comprises two different sources: Bachmair et al., 2016 and Stahl et al., 2016. It appears that our citation management software incorrectly displayed the first one as Bachmair, Stahl et al., 2016. We have rectified this issue in the revised manuscript.

Bachmair, S., Stahl, K., Collins, K., Hannaford, J., Acreman, M., Svoboda, M., Knutson, C., Smith, K. H., Wall, N., Fuchs, B., Crossman, N. D., & Overton, I. C. (2016). Drought indicators revisited: the need for a wider consideration of environment and society. WIREs Water, 3(4), 516-536. https://doi.org/https://doi.org/10.1002/wat2.1154

Stahl, K., Kohn, I., Blauhut, V., Urquijo, J., De Stefano, L., Acácio, V., Dias, S., Stagge, J. H., Tallaksen, L. M., Kampragou, E., Van Loon, A. F., Barker, L. J., Melsen, L. A., Bifulco, C., Musolino, D., de Carli, A., Massarutto, A., Assimacopoulos, D., & Van Lanen, H. A. J. (2016). Impacts of European drought events: insights from an international database of text-based reports. Nat. Hazards Earth Syst. Sci., 16(3), 801-819. https://doi.org/10.5194/nhess-16-801-2016

**P3L76**: Better to place the EDII and DIR references here. EDII: Stahl et al., 2016 and DIR: Smith et al., 2014?

Thank you for your suggestion. We agree that providing references to the EDII and DIR here would enhance the clarity of the manuscript. We have made the suggested change and addition in the revised version.

**P4L109**: I suggest to mention again the gap (instead of "that" gap) here since it is a new paragraph.

Thank you for the suggestion. In the revised manuscript, we have mentioned explicitly the gap again. We have replaced this sentence:

*"In this paper, the ambition was to fill that gap, with a focus on agricultural drought impacts at the national scale, across different crops and seasons, comparing the relative utility of traditional statistical methods at high resolution (remote sensing data at provincial scale) vs. lower resolution sectoral specific analyses (applying machine*

*learning approaches to regional/provincial yield data), to inform improved approaches for national DMEW."*

With the following two sentences:

*"The ambition of this paper was to fill the gap in the literature on studies investigating the links between drought indicators and impacts at a national scale in Thailand. Specifically, we focused on agricultural drought impacts, considering different crops and seasons, and compared the relative utility of traditional statistical methods at high resolution (i.e. remote sensing data at provincial scale) vs. lower resolution sectoral-specific analyses (i.e. applying machine learning approaches to regional/provincial yield data), to inform improved approaches for national DMEW."*

**P5L144**: Suggestion to rephrase the sentence: "….of Thailand. Although it has suffered….decades, there has been some…"

Thank you for the helpful suggestion. We agree that the suggested modification enhances the paragraph's readability. We have made this change in the revised manuscript.

**P8**: Figure 2. Here the authors clearly indicated that the RF model can be used to predict the crop yield (my general comment 1). This is one thing that I miss from the result.

As highlighted in our response to your general comment 1, we have used RF models primarily to study feature importance to better understand the link between drought indicators and drought impacts. However, we recognise the importance of using these models for impact prediction as well. To empathise this more, we have modified the first paragraph of the "Future Work" section (line 511-514) as follows:

*"In this study, we used RF models primarily to analyse the relationships between drought indicators and impacts, and to identify the relative importance and timing of relevant indicators for impacts on crops and forests. While the main focus of our analysis was on feature importance, our analysis also demonstrated the potential of RFs to simulate unseen data, which suggests they could be used for impact prediction. With further work, such as addressing the limitations discussed above, these models could be used for DMEWSs, support and compensation schemes, long-term planning, etc."*

**P9L8**: Please elaborate more how the authors did "detrended". The authors only said using a simple linear regression.

Thank you for highlighting the need to elaborate the way detrending was performed in our study. We have clarified on lines 208-210 and 227 that variables were

regressed against time using linear regression, and the residuals used to remove linear trends in the data. These are the edits in our manuscript:

Lines 208-210:

Before: "*The monthly crop-masked VCI was detrended before the correlation analysis (using a simple linear regression) in order to remove long-term trends, accounting for increased biomass from developments in agricultural technology and practices.*"

After: "*Monthly crop-masked VCI values were regressed against time using linear regression, and the residuals used to remove linear trends, accounting for increased biomass from developments in agricultural technology and practices.*"

Line 227:

Before: "*All input data were first de-trended using a simple linear regression*"

After: "*All input data were first regressed against time, and the residuals used as input to the random forest models to account for linear trends.*"

**P10L217**: "…both "indices" in our…….

Thank you for the correction, this has been added in the revised manuscript.

**P14L291**: I am wondering, it is VI or VCI?

Thank you for pointing out this potential confusion. To clarify, VI stands for Vegetation Index (as noted in line 90), while VCI stands for Vegetation Condition Index, which is one of the VIs we used in this study. The statement on line 291, "*For crops, we find high correlations between VI and SPEI of relatively short accumulation period during the dry season*", applies to both VIs considered in this study (VCI shown in the main manuscript, and VHI shown in the supplementary material). However, to avoid further confusion, we have revised the manuscript to only mention VCI in this sentence.

**P14L294-296**: Maybe elaborate more about the meaning of positive and negative correlations between VCI and meteo indicators. Also, the authors stated that short droughts are beneficial for forest growth. In my opinion, drought is never beneficial for any ecosystem. I suggest to rephrase the word beneficial.

Thank you for bringing up the need to elaborate on the meaning of positive and negative correlations between VCI and meteorological indicators and to rephrase our use of the word 'beneficial'. We agree with your points and have provided further clarification in the revised version. Specifically, we have added the following text:

*"A positive correlation between VCI and the meteorological indicators suggests that a deficit in water availability (as indicated by negative SPI or SPEI) leads to a decline in vegetation growth (reduced VCI). In contrast, a negative correlation suggests that such a deficit leads to an increase in vegetation growth. This second scenario may seem counterintuitive, but it can occur in energy-limited environments where water is not the limiting factor. In such cases, short duration droughts (i.e., periods drier than usual for the time of year) can stimulate increased vegetation growth, as droughts in energy-limited environments are often associated with increased radiation (i.e. energy) due to decreased cloud cover. This is discussed further in section 4.3."*

**P14L297**: The authors can consider to re-write "...the accumulation period best correlated is..." as "...the best correlated accumulation period is..."

Thank you for the suggestion. We have changed this in the revised version.

**P14L298**: Here and also in the discussion, the authors conclude that forest is more resistant to short droughts. I believe that this strongly relates to the ability of forest to subtract water from deeper layers, e.g. groundwater. Discuss this.

Thank you for bringing this to our attention. We agree that the deeper root systems of forest trees allow them to extract water from deeper layers of soil, making them more resilient to droughts compared to most crops. However, we have overlooked this point in our manuscript. To address this, we have included this explanation in the discussion section (4.3), along with references such as Breda et al. (2006) and Schenk and Jackson (2002), which support the role of deep roots in conferring resilience to droughts in forests. This is the new paragraph: "*Generally, the indicator showing the highest correlation with impacts is for longer accumulation period for forests than for crops, suggesting that shorter droughts will have impacts on crops whereas only longer droughts will affect forests. The higher resilience to droughts of forests compared to crops is at least partially explained by the deeper root systems of forest trees allowing them to extract water from deeper layers of the soil (Bréda et al., 2006; Schenk & Jackson, 2002).*"

Breda, N.; Huc, R.; Granier, A.; Dreyer, E. 2006. Temperate forest trees and stands under severe drought: a review of ecophysiological responses, adaptation processes and long-term consequences. Annals of Forest Science. 63(6): 625-644.

Schenk, H.J.; Jackson, R.B. 2002. The global biogeography of roots. Ecological Monographs. 72(3): 311-328.

**P18L343-346**: How to see the SPI24 from Figure 9 and to see 11 SPEI, 10 VCI, and 6 TCI from Figure 11? See my general comment 2.

We hope that the revised versions of Fig. 8-10 have addressed this issue.

**P22L397**: The authors stated that SPI is more important than SPEI. I am wondering whether the low precipitation in the N region has something to do with the result.

Thank you for your comment. This is an interesting point. The low precipitation in region N leads to the Actual Evapotranspiration (AET) to be water limited (i.e. AET < PET) and therefore SPEI could be less closely linked to agrometeorological conditions. However, this is also the case for region W and C, and to lesser extend NE as well, where precipitation is also low, but in these other regions, SPEI's importance is mostly dominant (Fig. 5c). We believe the dominant importance of SPI in region N is linked to the reliance of water storage for irrigation in this region, particularly for Corn_S2 which is planted in the dry season and relies heavily on irrigation (Fig. 10). Therefore, a deficit in rainfall (and consequent depleted storage) will have a strong impact on crop yield. We have added these reflections in the revised manuscript: "*This last point could be explained by the fact that low precipitation in region N leads to the Actual Evapotranspiration (AET) to be water limited (i.e. AET < PET) meaning SPEI may be less closely linked to agrometeorological conditions. However, this is also the case for region W and C, and to lesser extend NE as well, where precipitation is also low, but in these other regions, SPEI's importance is generally dominant (Fig.5c). Therefore, it is likely that the dominant importance of SPI in region N is linked to the reliance of water storage for irrigation in this region, particularly for Corn S2 which is planted in the dry season and relies heavily on irrigation (Fig.10). Therefore, a deficit in rainfall (and consequent depleted storage) will have a strong impact on crop yield.*"

**P22L410-411**: You may discuss the difference in water consumption by each crop.

Thank you for your comment. We appreciate your suggestion to discuss the difference in water consumption by each crop. The irrigation requirement indeed varies greatly between crops. Paddy rice is the most water-intensive crop, with an irrigation requirement of around 520m$^3$/ton if cultivated during the wet season and 1140m$^3$/ton in the dry season. Corn, on the other hand, requires irrigation only if cultivated in the dry season, with an irrigation requirement of approximately 850m$^3$/ton. Finally, cassava is the least water-demanding crop, with an irrigation requirement of around 20m$^3$/ton in the wet season and 65m$^3$/ton in the dry season (Gheewala et al., 2014).

We have included these numbers in our revised manuscript to provide a more comprehensive understanding of the impact of drought on crop yields. Thank you for this helpful suggestion. This is the paragraph that has been added to section 4.1: "*Also, Cassava is the least water-demanding crop of the list (irrigation requirement of around 20m3/ton in the wet season and 65m3/ton in the dry season, Gheewala et al., 2014). This explains the comparatively lower importance of long accumulation indicators for Cassava (Fig.10), given less reliance on water storage, especially compared with the most water-intensive crops, such as paddy rice (irrigation requirement of 520m3/ton*

*during the wet season and 1140m3/ton in the dry season) and Corn S2 (irrigation requirement of 850m3/ton in the dry season)."*

**Reference:**

Gheewala, S.H.; Silalertruksa, T.; Nilsalab, P.; Mungkung, R.; Perret, S.R.; Chaiyawannakarn, N. Water Footprint and Impact of Water Consumption for Food, Feed, Fuel Crops Production in Thailand. Water 2014, 6, 1698-1718. https://doi.org/10.3390/w6061698

**P23L437**: Rephrase "though this effect is highly variety specific:

This has been rephrased as: *"However, it should be noted that this effect varies significantly depending on the specific crop variety, [...]"*

**P24L453-455**: Make two sentences.

Thank you for the suggestion. We agree that the sentence is too long and we have splitted it into two sentences as follows:

*"For the crops where it was possible to build a RF model, the analysis of the temporal variation in feature importance and the indicator-to-impact relationships provide insights into critical periods of the year for early warning of impacts and relevant accumulation period. Specifically, these are periods of interest when dry conditions could lead to impacts."*

**P24L459**: "…seasons, which suggests… -> "…seasons, suggest…"

Thank you, this has been changed in the revised manuscript.

**P24L468-470**: Explain this already in the beginning, thus the readers will not be confused.

We have mentioned this as a response to your comment on **P14L294-296** earlier. Hopefully this addresses this point too.

**P25L498**: Here, the authors can link the short drought events with the limitation of using data-driven model, such as machine learning.

Thank you for your comment. To clarify, did you mean 'short drought events' or 'short period of record'? Assuming you meant the latter, we agree that the limitation of using data-driven models such as machine learning is the need for a large amount of data to train the model effectively. In our study, we had a relatively short period of record, which limited the amount of data available for training the models. As a result, the models may not have been able to accurately capture the full range

of conditions that could occur in the real world. For example, for species such as longan, which are more susceptible to long drought events, the limited instances of these events in our training data may have affected the model's ability to accurately predict impacts. We have added the following paragraph to section 4.4 of the discussion in the revised manuscript: "*Lastly, another limitation of using data-driven models such as RFs is the need for a large amount of data needed to train the model effectively. In our study, we had a relatively short period of data available, which limited the amount of data available for training the models. As a result, the models may not have been able to accurately capture the full range of conditions that could occur in the real world. For example, for species such as longan, which are more susceptible to long drought events, the limited instances of these events in our training data may have affected the model's ability to accurately predict impacts.*"

**P25L505-506**: Rephrase "Though powerful tools to produce predictive models from data"

We have rephrased this sentence as follows:

"*RFs are powerful tools for producing predictive models from data, but they are considered 'black boxes' since they do not explicitly extract the relationships between input features and the predicted outcomes. However, RFs can aid in the interpretation of the model through the analysis of feature importance, which identifies the most influential variables in making predictions.*"

**References:**

Hariadi et al. (2023). A high-resolution perspective of extreme rainfall and river flow under extreme climate change in Southeast Asia, https://doi.org/10.5194/hess-2023-14.

Smith et al. (2014). Local observers fill in the details on drought impact reporter maps, https://doi.org/10.1175/1520-0477-95.11.1659.

==================================================================

We would like to thank Veit for taking the time to review our manuscript and for providing insightful and constructive comments, which have contributed to strengthen our manuscript.

Below is a point-by-point response to all comments. Original comments are in **black**, whereas the authors' responses are in **blue**, and actual changes in the manuscript are shown in **red**.

Dear authors,

First of all I have to apologise for the strong delay in reviewing your paper. Overall it was a pleasure to review your excellent paper. Very well written and designed in a carefully thought-out way. I believe this study to be essential to support agricultural drought management in the future.

Thank you very much for your positive feedback and kind words regarding our manuscript. We appreciate the time and effort you have invested in reviewing our work, and we understand that the delay in the review process can be unavoidable. We are grateful for your thorough evaluation of our study and for recognising its significance in supporting agricultural drought management. Once again, thank you for your valuable contribution to our research.

In addition to the comments of SJ Sutantos comments there is only few to add on. All over I recommend this manuscript to be published after minor revisions.

I would appreciate if you could add some introductionary thoughts on the usage of the vegetation indices and their classification it they are rather used a drought index or as a proxy for impacts/ impacts. Also, if VHI can be used without any knowledge on the hazard situation?

Thank you for drawing our attention to the need for introductory thoughts on vegetation indices in our manuscript. We agree that this was missing, and in response, we have included the following text in our introduction on the usage of vegetation indices in the revised manuscript. We have removed any duplicate information from the Data section (2.2).

*VIs are commonly used to monitor the impacts of drought on vegetation. The Normalised Difference Vegetation Index (NDVI) is one of the most established and widely used VIs (Tucker, 1979). It exploits the sharp increase in vegetation reflectance across the red and*

*near-infrared (NIR) regions of the electromagnetic spectrum, known as the 'red-edge', to detect photosynthetically active plant material and infer plant stress. However, the Vegetation Condition Index (VCI), a pixel-based normalization of NDVI, offers a more robust indicator for seasonal droughts by minimising spurious or short-term signals and amplifying long-term trends (Anyamba & Tucker, 2012; Liu & Kogan, 1996). VCI has been widely used and has proved to be effective in monitoring vegetation change and signalling agricultural drought (e.g. Jiao et al., 2016). The Vegetation Health Index (VHI) is a composite index that combines the VCI and Temperature Condition Index (TCI) – a pixel-based normalisation of the Land Surface Temperature (LST) – and is also commonly used to monitor vegetation stress and drought conditions (Kogan, 1997). VHI incorporates the effect of temperature and is therefore more suitable for monitoring the effect of drought in species more sensitive to concurrent water and heat stress. VHI has been successfully used worldwide to monitor vegetation stress and drought conditions (e.g. Jain et al., 2009; Singh et al., 2003; Unganai & Kogan, 1998). Note that these VIs are relative indices that compare current conditions to the long-term average to measure vegetation health, and therefore are dependent on the environmental and climatic conditions of the study area. As such, they should be used in conjunction with information on the drought hazard situation to distinguish between drought and different hazards on vegetation (e.g. disease, floods, anthropogenic impacts, etc.).*

*In addition to their use as drought indicators as discussed above, VIs are often used as proxies for agricultural drought impacts. The relationship between crop yield and VIs varies by crop type and location but has been shown to be strong in many locations. For example, strong correlations were found between VIs and crop yield in North America (e.g. maize in Bolton & Friedl, 2013; winter wheat, sorghum and corn in Kogan et al., 2012), South America (e.g., white oat in Brazil in Coelho et al., 2020), Europe (e.g., maize in Germany in Bachmair et al., 2018; cereals in Spain in García-León et al., 2019), Asia (e.g., sugarcane in India in Dubey et al., 2018), the Middle East (e.g., paddy rice in Iran in Shams Esfandabadi et al., 2022), Africa (e.g., millet and sorghum in the Sahelian region in Maselli et al., 2000), and Australia (e.g., wheat in Smith et al., 1995).*

**References:**

Anyamba, A. and C.J. Tucker, Historical perspectives on AVHRR NDVI and vegetation drought monitoring. B.D. Wardlow, M.C. Anderson, J.P. Verdin (Eds.), Remote Sensing of Drought: Innovative Monitoring Approaches, CRC Press, New York, United States of America, 2012: p. 23-51. DOI: 10.1201/b11863.

Bachmair, S., M. Tanguy, J. Hannaford and K. Stahl, How well do meteorological indicators represent agricultural and forest drought across Europe? Environmental Research Letters, 2018. 13(3): p. 034042. DOI: 10.1088/1748-9326/aaafda.

Bolton, D.K. and M.A. Friedl, Forecasting crop yield using remotely sensed vegetation indices and crop phenology metrics. Agricultural and Forest Meteorology, 2013. 173: p. 74-84. DOI: https://doi.org/10.1016/j.agrformet.2013.01.007.

Coelho, A.P., R.T. de Faria, F.T. Leal, J.d.A. Barbosa and D.L. Rosalen, Validation of white oat yield estimation models using vegetation indices. Basic areas, Bragantia 79 (2), 2020. DOI: 10.1590/1678-4499.20190387.

Dubey, S.K., A.S. Gavli, S.K. Yadav, S. Sehgal and S.S. Ray, Remote Sensing-Based Yield Forecasting for Sugarcane (Saccharum officinarum L.) Crop in India. Journal of the Indian Society of Remote Sensing, 2018. 46(11): p. 1823-1833. DOI: 10.1007/s12524-018-0839-2.

Esfandabadi, H.S., M.G. Asl, Z.S. Esfandabadi, S. Gautam and M. Ranjbari, Drought assessment in paddy rice fields using remote sensing technology towards achieving food security and SDG2. British Food Journal, 2022. 124(12): p. 4219-4233. DOI: 10.1108/BFJ-08-2021-0872.

García-León, D., S. Contreras and J. Hunink, Comparison of meteorological and satellite-based drought indices as yield predictors of Spanish cereals. Agricultural Water Management, 2019. 213: p. 388-396. DOI: https://doi.org/10.1016/j.agwat.2018.10.030.

Jain, S.K., R. Keshri, A. Goswami, A. Sarkar and A. Chaudhry, Identification of drought-vulnerable areas using NOAA AVHRR data. International Journal of Remote Sensing, 2009. 30(10): p. 2653-2668. DOI: 10.1080/01431160802555788.

Jiao, W., L. Zhang, Q. Chang, D. Fu, Y. Cen and Q. Tong, Evaluating an Enhanced Vegetation Condition Index (VCI) Based on VIUPD for Drought Monitoring in the Continental United States. Remote Sensing, 2016. 8(3): p. 224.

Kogan, F.N., Global Drought Watch from Space. Bulletin of the American Meteorological Society, 1997. 78(4): p. 621-636. DOI: 10.1175/1520-0477(1997)078<0621:GDWFS>2.0.CO;2.

Kogan, F., L. Salazar and L. Roytman, Forecasting crop production using satellite-based vegetation health indices in Kansas, USA. International Journal of Remote Sensing, 2012. 33(9): p. 2798-2814. DOI: 10.1080/01431161.2011.621464.

Liu, W.T. and F.N. Kogan, Monitoring regional drought using the Vegetation Condition Index. International Journal of Remote Sensing, 1996. 17(14): p. 2761-2782. DOI: 10.1080/01431169608949106.

Maselli, F., S. Romanelli, L. Bottai and G. Maracchi, Processing of GAC NDVI data for yield forecasting in the Sahelian region. International Journal of Remote Sensing, 2000. 21(18): p. 3509-3523. DOI: 10.1080/014311600750037525.

Singh, R.P., S. Roy and F. Kogan, Vegetation and temperature condition indices from NOAA AVHRR data for drought monitoring over India. International Journal of Remote Sensing, 2003. 24(22): p. 4393-4402. DOI: 10.1080/0143116031000084323.

Smith, R.C.G., J. Adams, D.J. Stephens and P.T. Hick, Forecasting wheat yield in a Mediterranean-type environment from the NOAA satellite. Australian Journal of Agricultural Research, 1995. 46(1): p. 113-125.

Tucker, C.J., Red and photographic infrared linear combinations for monitoring vegetation. Remote Sensing of Environment, 1979. 8(2): p. 127-150. DOI: https://doi.org/10.1016/0034-4257(79)90013-0.

Unganai, L.S. and F.N. Kogan, Southern Africa's recent droughts from space. Advances in Space Research, 1998. 21(3): p. 507-511. DOI: https://doi.org/10.1016/S0273-1177(97)00888-0.

In your methods you mention that you spatially aggregated the standardised indices. Please elaborate on your practise applied with a focus on a) the spatial aggregation method of drought indices, were the standardised indices aggregated to province levels or the indicators (temp, precip.) and then the distribution done? And b) how did you aggregate the indices in time.

Thank you for bringing up the need for more details on how the aggregation was carried out in our study. We have added additional information on this aspect in the revised manuscript, with a specific focus on spatial aggregation. To address your questions:

a) Spatial aggregation:

To conduct the correlation analysis, we spatially averaged the meteorological variables (precipitation and PET) for each province and then calculated the standardised indicators based on the province-averaged time series. For vegetation indices, we first derived them at the pixel level for the entire country, and then used a land cover map to differentiate between forest and crop-covered pixels. We then calculated province-level vegetation indices averages separately for forest and crops, using the corresponding land cover mask.

Regarding the random forest modelling, we used the same data as for the correlation analysis, but grouped all the data belonging to the provinces included in each of the 6 regions. This was done because there was not enough data to train machine learning models at the provincial level.

b) Temporal aggregation:

We used monthly time series for most of our analyses, with the only exception being when we compared vegetation indices with crop yield data to validate the use of VIs as proxies for drought impact. In this case, we averaged the VIs over the growing season for each crop, as explained in lines 197-200 of the manuscript.

We hope that these additional details have addressed your concerns and provide a better understanding of the methods used in our study. This is the paragraph that has been added to our revised manuscript:

***Spatial and temporal aggregation***

*To derive the meteorological indicators, we first averaged the meteorological variables (precipitation and PET) for each province and then calculated the standardised indicators based on the province-averaged time series. For VIs, we first derived them at the pixel level for the entire country, and then used a land cover map to differentiate between forest and crop-covered pixels. We then calculated province-level VIs averages separately for forest and crops, using the corresponding land cover mask. We used monthly time series for most of our analysis, with the exception of the comparative analysis between VIs and crop yield (described further in section 2.3.1.1) where VIs were averaged over the growing season for each crop.*

In my opinion, the discussion on your initial analysis (Fig 4) is a little short and could tolerate a little more discussion on possible drivers (maybe in the discussion section and not in the results). In figure 4b) East inland Thailand, there are three regions neighbouring, having the same major crops paddy rice (and high percentages), but there are either VHI, VCI or negatively correlated.--> why do they perform so different? Irrigation practise (e.g. river fed irrigation?) Elevation?

We agree that this is an interesting question that could be further investigated. However, we have deliberately kept that discussion short, as we conducted this analysis as a test to ensure that using vegetation indices as proxies for drought impacts was a reasonable assumption for the rest of our (main) analysis. Based on our results, we are generally satisfied with this assumption. However, there are some exceptions where neighbouring provinces with similar land cover, climatology and dominant crop show very different correlations with VIs, as you note in the case of some eastern provinces in Thailand. It is conceivable that differences in irrigation or agricultural practices, or in the outbreak of pests and diseases, could be contributing factors, but we do not have any evidence to support these hypotheses. Therefore, we have chosen not to delve further into this topic in the present study. However, we have highlighted this gap in our revised manuscript and added a sentence as follows: '*In some cases, there is no obvious reason as to why the correlation is very different between two neighbouring provinces which share similar topography, land cover, climatology and dominant crop type. However, differences in irrigation or*

*agricultural practices, or in the outbreak of pests and diseases, could be contributing factors. Exploring these factors in future research may provide insights into the observed differences in correlations.'*

Some minor points:

Figure 1-4 – Names of neighbouring countries are not readable

We thank you for bringing to our attention the issue with the readability of neighbouring countries' names in these maps. We have tried to address this concern in the revised version. However, we would like to inform you that there are limitations to the modifications we can make to the background layer since we used ESRI Basemap, which has pre-set formatting and display for its layers. Nevertheless, we have made an effort to increase the font size of the country names by resizing the images. See for example the amended Figure 1:

[Figure]

Figure 4+ please increase legend size

The legend for these maps have been made larger in the revised version.

Please find the new Figures 4-6 in the following pages.

New Figure 4:

[Figure]

New Figure 5:

[Figure]

New Figure 6:

[Figure]

Furthermore, you might check on the following literature. Their results might be useful for some discussion and or introduction.

Sa-Nguansilp, C., Wijitkosum, S., Sriprachote, A., 2017. Agricultural drought risk assessment in Lam Ta Kong Watershed, Thailand. International Journal of Geoinformatics 13 (4), 37–43.

Monkolsawat, C., et al., 2001. An. Evaluation of Drought Risk Area in NE Thailand Asian Journal of Geoinformatics 1 (4), 33–44.

Wijitkosum S 2018. Fuzzy AHP for drought risk assessment in lam Ta Kong watershed, the north-eastern region of Thailand. Soil and Water Research, 13(4), 218–225. doi:10. 17221/158/2017-SWR

Thank you for bringing these references to our attention. We have reviewed them and believe they are relevant to our study, particularly in highlighting the vulnerability of the Northeast of Thailand to agricultural drought. We have cited these references in our introduction to provide additional context. The following sentence was added to the paragraph on regional differences in section 2.1: "*This region is the most prone to drought (LePoer, 1987), and as such, is particularly vulnerable to agricultural droughts as highlighted by several studies (e.g. Mongkolsawat et al., 2001; Sa-nguansilp et al., 2017; Wijitkosum, 2018).*"